# Helix 8 is the essential structural motif of mechanosensitive GPCRs

Serap Erdogmus[1,10], Ursula Storch[1,2,10], Laura Danner[1], Jasmin Becker[1], Michaela Winter[1], Nicole Ziegler[3], Angela Wirth[4,5], Stefan Offermanns[4,6], Carsten Hoffmann [7], Thomas Gudermann[1,8,9]* & Michael Mederos y Schnitzler[1,8]*

G-protein coupled receptors (GPCRs) are versatile cellular sensors for chemical stimuli, but also serve as mechanosensors involved in various (patho)physiological settings like vascular regulation, cardiac hypertrophy and preeclampsia. However, the molecular mechanisms underlying mechanically induced GPCR activation have remained elusive. Here we show that mechanosensitive histamine $H_1$ receptors ($H_1Rs$) are endothelial sensors of fluid shear stress and contribute to flow-induced vasodilation. At the molecular level, we observe that $H_1Rs$ undergo stimulus-specific patterns of conformational changes suggesting that mechanical forces and agonists induce distinct active receptor conformations. GPCRs lacking C-terminal helix 8 (H8) are not mechanosensitive, and transfer of H8 to non-responsive GPCRs confers, while removal of H8 precludes, mechanosensitivity. Moreover, disrupting H8 structural integrity by amino acid exchanges impairs mechanosensitivity. Altogether, H8 is the essential structural motif endowing GPCRs with mechanosensitivity. These findings provide a mechanistic basis for a better understanding of the roles of mechanosensitive GPCRs in (patho)physiology.

[1] Walther Straub Institute of Pharmacology and Toxicology, Ludwig Maximilian University of Munich, Goethestr. 33, 80336 Munich, Germany. [2] Institute for Cardiovascular Prevention (IPEK), Ludwig Maximilian University of Munich, Pettenkoferstr. 9, 80336 Munich, Germany. [3] Institute of Pharmacology and Toxicology, Julius Maximilian University of Würzburg, Versbacher Str. 9, 97078 Würzburg, Germany. [4] Max Planck Institute for Heart and Lung Research, Department of Pharmacology, Ludwigstraße 43, 61231 Bad Nauheim, Germany. [5] Institute of Pharmacology, University of Heidelberg, Im Neuenheimer Feld 366, 69120 Heidelberg, Germany. [6] J. W. Goethe University Frankfurt, Medical Faculty, 60590 Frankfurt, Germany. [7] Institute for Molecular Cell Biology, Center for Molecular Biomedicine, Friedrich Schiller University Jena, Hans Knoell Str. 2, 07745 Jena, Germany. [8] DZHK (German Centre for Cardiovascular Research), Munich Heart Alliance, Munich, Germany. [9] Comprehensive Pneumology Center Munich (CPC-M), German Center for Lung Research, Munich, Germany. [10]These authors contributed equally: Serap Erdogmus, Ursula Storch. *email: thomas.gudermann@lrz.uni-muenchen.de; mederos@lrz.uni-muenchen.de

GPCRs serve as molecular targets for about 30% of all approved drugs[1]. They are versatile cellular sensors activated not only by hormones and neurotransmitters, but also by physical and chemical stimuli such as voltage[2–6], ions[7], and mechanical forces[8,9]. Until now, several GPCRs like apelin receptors[10,11], sphingosine 1-phosphate receptors (S₁PR)[12], parathyroid hormone 1 receptors (PTH₁R)[13], dopamine D5 receptors (D₅R)[14], angiotensin II AT 1 receptors (AT₁R)[15–18], GPR68 receptors[19], cysteinyl leukotriene 1 receptors (CysLT₁R)[20], bradykinin B2 receptors (B₂R)[21], formyl peptide 1 receptors[22], endothelin ET$_A$ receptors[15], muscarinic M₅ receptors[15], and vasopressin V$_{1A}$ receptors[15] have been identified as mechanosensors involved in physiological settings like vascular regulation[15,16,18–20] as well as in pathophysiological circumstances like cardiac hypertrophy[17] and preeclampsia[23]. Thus, dissecting the structure–function relationship of mechanosensitive GPCRs is a crucial first step towards a deeper understanding of their roles in physiology and pathophysiology and might help improve pharmacotherapy.

It is well known that in blood vessels, mechanical forces can elicit biological responses[24] essential for autoregulatory vessel function. Increased blood pressure can activate smooth muscle cells resulting in myogenic vasoconstriction known as the Bayliss effect[25]. By contrast, blood flow causes shear stress that activates endothelial cells resulting in flow-induced vasodilation[26–29] thereby increasing vessel perfusion. Flow-induced vasodilation is disturbed in endothelial dysfunction resulting from pathophysiological states like atherosclerosis[30]. Shear stress increases the intracellular calcium concentration $[Ca^{2+}]_i$ in endothelial cells leading to $Ca^{2+}$/calmodulin-dependent activation of endothelial nitric oxide synthase (eNOS)[31–33] and to the release of nitric oxide (NO) responsible for vasodilation[32–35]. However, the molecular identity of endothelial mechanosensors is still a matter of debate.

A large body of evidence suggests the involvement of numerous proteins in mechanosensation and –transduction in endothelial cells. These include apical mechanosensors such as primary cilia, the glycocalyx, ion channels, GPCRs, receptor tyrosine kinases and caveloae, junctional mechanosensors such as platelet endothelial cell adhesion molecule-1 (PECAM-1), VE-Cadherin and VEGF receptors and basal sensors such as integrins (summarized in[36]). Among the mechanosensitive GPCRs, S₁PR[12], B₂R[21], and GPR68[19] are discussed as potential endothelial mechanosensory proteins. Moreover, ion channels like PIEZO1 might also act as mechanosensors. Recently, PIEZO1 has been identified as an endothelial sensor of shear stress that evokes ATP and adrenomedullin release causing GPCR and eNOS activation and resultant nitrogen oxide (NO) production thereby leading to vasodilation[37,38].

To extend the current knowledge about the physiological role of intrinsically mechanosensitive GPCRs, we focus on the mechanosensitive G$_{q/11}$-protein coupled H₁R[15,39] characterized by the most pronounced mechanosensitivity of any GPCR tested by us[15]. H₁Rs are highly expressed in the endothelium, with higher expression levels in endothelial than in smooth muscle cells[40]. However, a potential physiological role of H₁Rs as endothelial mechanosensors has not been investigated yet. We find that endogenously expressed H₁Rs in endothelial cells are sensitive both to shear stress and to membrane stretch. Therefore, we set out to elucidate the molecular principles underlying mechanosensation of H₁Rs in particular and of GPCRs in general.

Structural analysis of GPCRs[41] has shown that agonist stimulation mainly causes conformational changes of transmembrane domain (TM) 6, and applying a substituted cysteine accessibility mapping provided evidence that mechanical GPCR stimulation might induce conformational changes of TM7[42]. However, the dynamics of mechanically induced conformational changes and the structural motifs mediating mechanosensation in GPCRs have not been identified yet. Indeed, it is not even clear whether mechanical force acts directly or indirectly on these GPCRs.

To address these questions on a molecular level, we employ the technique of intramolecular dynamic fluorescence energy transfer (FRET) to monitor conformation changes of H₁R in order to analyze whether mechanical forces and agonists foster distinct active receptor conformations. In addition, we aim to identify molecular structures that are essential for mechanosensation of GPCRs.

We find that the H₁R undergoes stimulus-specific patterns of conformational changes, and we identify the C-terminal helix 8 (H8) as the essential structural motif endowing H₁R and other GPCRs with mechanosensitivity. On the physiological level, we identify H₁R as a sensor of fluid shear stress in the endothelium contributing to flow-induced vasodilation.

## Results

**The endothelial H₁R is a sensor of shear stress.** To investigate a potential physiological role for the mechanosensitive H₁R in flow-induced endothelial stimulation, we first investigated its expression levels in primary endothelial cells derived from human umbilical veins (HUVEC) serving as a cell model to analyze the mechanosensitivity of endogenously expressed H₁Rs. In HUVEC, H₁R is more abundantly expressed than any other GPCR that we tested (Fig. 1a). H₁R showed more than 12-fold higher mRNA expression levels than other G$_{q/11}$-protein coupled receptors and more than 4-fold higher levels than G$_s$-protein coupled β-adrenergic receptors (β$_x$R). mRNA expression levels of other mechanosensitive GPCRs like V$_{1A}$R, CysLT₁R, GPR68, D₅R and PTH₁R were below the detection limit. mRNA expression of the mechanosensitive AT₁R was more than 400-fold lower than that of H₁R suggesting a negligible role of AT₁R for mechanosensation in HUVECs.

We then performed calcium imaging with HUVEC. Shear stress of 4 and 20 dyn cm$^{-2}$ induced calcium transients that were significantly, but not fully suppressed by the selective inverse H₁R agonist mepyramine (Fig. 1b, c). Endothelial H₁R was also sensitive to hypoosmotic membrane stretch induced by short-time application (≤60 s) of a hypoosmotic solution[15] (Fig. 1d, e) which was used as a different mechanical stimulus. Hypoosmotic membrane stretch similarly caused calcium transients in HEK293 cells heterologously overexpressing H₁R (Fig. 1f, g). Mepyramine nearly completely abolished hypoosmotically induced calcium transients (Fig. 1d, e), indicating that H₁Rs were responsible for these calcium responses. Thus, endogenously expressed H₁Rs are sensitive both to membrane stretch and to shear stress.

Since HUVEC are of premature nature and not fully differentiated, we next verified our results in a physiological setting by analyzing flow-induced vasodilation of isolated murine mesenteric artery segments. To test whether endothelial H₁R might be involved in flow-induced vasodilation of conduit arteries, mesenteric artery segments from mice were pre-constricted up to 20% either with the thromboxane A₂ receptor agonist U46619 (Fig. 1h) or with a bath solution containing 35 mM potassium chloride (Fig. 1i). In wild-type (C57BL/6 J) arteries, application of intravascular shear stress of 4.8 ± 0.5 (mean ± sem) and 8.8 ± 1.1 dyn cm$^{-2}$ (mean ± sem) resulted in increasing vasodilation (Fig. 1h, i) that was significantly suppressed by the inverse H₁R agonists mepyramine or desloratadine. In arteries from H₁R (H₁R$^{-/-}$)[43] and H$_{1/2/3/4}$R quadruple gene-deficient mice (H$_{1/2/3/4}$R$^{-/-}$)[44] vasodilation was significantly diminished (Fig. 1h, i and Supplementary Fig. 1). There were no significant differences between H₁R$^{-/-}$,

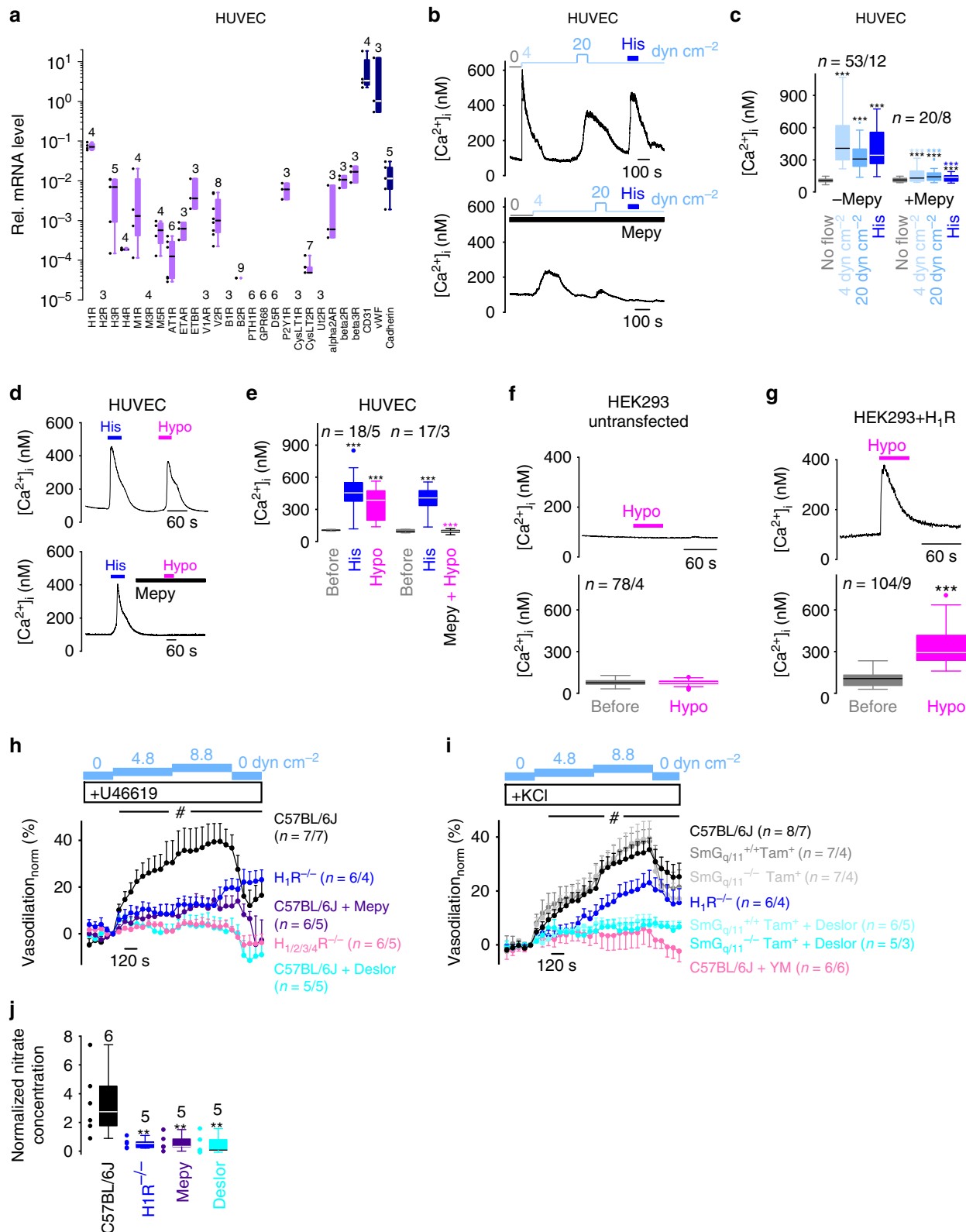

$H_{1/2/3/4}R^{-/-}$ or between mepyramine- or desloratadine-treated arteries. To investigate whether shear stress-induced vasodilation involves NO production, we measured nitrate concentrations in vessel perfusates that were collected during vasodilation experiments. Nitrate concentrations in vessel perfusates from $H_1R^{-/-}$ and from wild-type arteries treated with mepyramine or desloratadine were strongly reduced (Fig. 1j). These findings

suggest that shear stress activates $H_1R$ resulting in endothelial $Ca^{2+}$ transients and subsequent NO production.

To rule out an involvement of $H_1R$ expressed in vascular smooth muscle cells, we next analyzed arteries from smooth muscle-specific $G_{q/11}$-protein knock-down mice ($SmG_{q/11}^{-/-}$)[45]. There was no difference between arteries from wild-type and from $SmG_{q/11}^{-/-}$ or wild-type littermates ($SmG_{q/11}^{+/+}$), which

**Fig. 1 The endothelial $H_1R$ is a sensor of shear stress. a** mRNA expression of GPCRs and endothelial markers in HUVEC. CD31 cluster of differentiation 31, vWF von Willebrand factor, Cadherin vascular endothelial cadherin. Numbers indicate numbers of independent experiments. **b–g** Calcium imaging measurements. **b, d, f, g** Representative traces of $[Ca^{2+}]_i$. Applications of shear stress, 100 µM histamine (His), 30 µM (**d**) or 100 µM (**b**) mepyramine (Mepy) and hypoosmotic solution (Hypo, 150 mOsm kg$^{-1}$). **c, e–g** Summaries of $[Ca^{2+}]_i$. (**c**) ***$P < 0.001$, black asterisks; Wilcoxon matched-pairs signed-rank test to compare to no flow conditions and ***$P < 0.001$, blue asterisks; Mann–Whitney $U$ test to compare $[Ca^{2+}]_i$ in the presence or absence of mepyramine. **e** ***$P < 0.001$, black asterisks; Wilcoxon matched-pairs signed-rank test to compare to basal $[Ca^{2+}]_i$, ***$P < 0.001$, magenta asterisks; Mann–Whitney $U$ test to compare hypoosmotically induced $[Ca^{2+}]_i$ signals in the presence and absence of mepyramine. **f, g** ***$P < 0.001$; Wilcoxon matched-pairs signed-rank test to compare to basal $[Ca^{2+}]_i$. **h, i** Summaries of flow-induced vasodilations with murine mesenteric artery segments. Wild-type arteries (C57BL/6 J) without and with incubation of 100 µM mepyramine (Mepy), 30 µM desloratadine (Deslor) or 100 nM YM254890 (YM). Arteries from $H_1R^{-/-}$ and $H_{1/2/3/4}R^{-/-}$ mice. Tamoxifen-induced, smooth muscle-specific $G_{q/11}$-protein knock-down arteries (SmG$_{q/11}^{-/-}$Tam$^+$) and tamoxifen-treated WT littermates (SmG$_{q/11}^{+/+}$Tam$^+$) in the presence and absence of 30 µM desloratadine. $n$ indicates the number of arteries and the number of mice. Pre-constriction with 20 nM U46619 (**h**) or 35 mM KCL (**i**). #$P < 0.05$ from minute 8. Kruskal–Wallis test. Data are displayed as mean ± sem. **j** Summary of normalized nitrate concentrations in vessel perfusates from indicated arteries in the presence and absence of mepyramine or desloratadine. $n$ indicates the number of independent experiments. **$P < 0.01$; Mann–Whitney $U$ test compared to C57BL/6 J. **c, e–g** $n = x/y$ indicates the sample size, where $x$ is the number of measured **c**ells and $y$ is the number of coverslips from at least 3 experimental days. **a, c, e–g, j** Data are presented as boxplots (median plus interquartile range (IQR) and whiskers (max. 1.5-fold IQR)). See also Supplementary Fig. 1. Source data are provided as a Source Data file.

served as a controls (Fig. 1i and Supplementary Fig. 1). The $G_{q/11}$-protein inhibitor YM254890[46] also abolished flow-induced vasodilation (Fig. 1i and Supplementary Fig. 1). Vessel parameters like outer diameters at no flow conditions, maximal 60 mM KCl-induced vasoconstriction, acetylcholine-induced vasodilation and maximal vasodilation induced by $Ca^{2+}$-free solutions at intraluminal pressures of 50 and 120 mmHg were not different between the genotypes (Supplementary Fig. 1) indicating that the vessels were comparable. Histamine was not detectable in vessel perfusates using a commercially available enzyme immunoassay with a detection limit of 1.8 nM suggesting that endothelial $H_1R$ activation by shear stress was agonist-independent. To summarize, our findings support the concept that $H_1R$ is activated by shear stress resulting in to $G_{q/11}$-protein activation and subsequent NO production. This signaling pathway significantly contributes to flow-induced vasodilation in mesenteric arteries.

**$H_1R$ adopts distinct mechanically induced conformations.** To analyze the molecular mechanism underlying mechanosensation of $H_1Rs$, we applied the method of dynamic intramolecular fluorescence resonance energy transfer (FRET) to HEK293 cells expressing recombinant $H_1R$. First, we aimed to differentiate between mechanically versus agonist-induced conformational changes of the $H_1R$. To this end, we started out with three different master $H_1R$ constructs by attaching cerulean as the FRET donor to the C-terminus and by inserting binding motifs for the fluorescein arsenical hairpin binder FlAsH (as the FRET acceptor) into three different positions of the receptor: two into the third intracellular loop (at the beginning: $H_1R$-il3-b; at the end: $H_1R$-il3-e) and one at a proximal position of the C-terminus just in front of the C-terminal H8 ($H_1R$-ct-b) (Fig. 2a). H8 is a common feature of several GPCRs. All $H_1R$ constructs similarly responded to agonist stimulation monitored by calcium imaging, and they decorated the plasma membrane as assessed by confocal microscopy (Supplementary Fig. 2). Maximal dimercaprol (BAL)-induced FRET reductions and maximal FRET efficiencies of all constructs did not differ, indicating that FRET changes of the constructs are directly comparable (Supplementary Fig. 3). To additionally test whether the kinetics of agonist-induced FRET signal changes is comparable to the published muscarinic $M_3$ receptor ($M_3R$)[47], we compared agonist-induced conformational changes. In isosmotic solutions with a reduced NaCl concentration the kinetics of agonist-induced FRET signals was $\tau_{1/2} = 251 \pm 18$ ms (mean ± sem), in a physiological bath solution $\tau_{1/2} = 148 \pm 7$ ms (mean ± sem). The kinetics of membrane stretch-induced FRET signal decreases was characterized by a $\tau_{1/2}$ value

of $381 \pm 32$ ms (mean ± sem). Thus, the conformational kinetics of the $H_1R$ was slightly slower (2.4-fold), but in the same range as reported for the $M_3R$[47] (Supplementary Fig. 2k, l).

Mechanical stimulation by applying hypoosmotic membrane stretch of FlAsH-labeled $H_1R$ constructs caused significantly larger reductions of FRET signals than histamine at maximally effective concentrations, independently of the order of stimulus application (Fig. 2b–e and Supplementary Fig. 4a, b). The $H_1R$-ct-b variant did not display any agonist-induced FRET changes, but responded most robustly to membrane stretch (Fig. 2e). Since both fluorophores frame H8 of the $H_1R$-ct-b construct, the observed FRET signal decreases may originate from increasing distances between the two fluorophores and might therefore reflect an elongation of H8. The inverse $H_1R$ agonist mepyramine (30 µM) significantly reduced mechanically induced FRET signals of all $H_1R$-FRET constructs, and the even more selective levocetirizine (10 µM) significantly reduced FRET signals of the $H_1R$-il3-e and of the $H_1R$-ct-b, but not of the $H_1R$-il3-b construct (Fig. 2f). However, both inverse agonists almost fully suppressed hypoosmotically induced calcium transients of all $H_1R$-FRET constructs (Supplementary Fig. 4c-e). These findings suggest that the FRET signals elicited by membrane stretch in the presence of the inverse agonists reflect inactive receptor conformations that do not allow for G-protein activation and subsequent signaling.

Analyzing the $H_1R$-il3-b construct which had shown the smallest mechanically induced FRET signals we still observed a correlation between increasing hypoosmotic stimulations (Fig. 2g, h) and the resultant FRET signals. Increasing fluid shear stress (Fig. 2i, j) which was used as a different mechanical stimulus had the same effect entailed stepwise decreasing FRET signals thus meeting a central criterion of a mechanosensor[9]. Shear stress-induced FRET signals of the $H_1R$-il3-b and the $H_1R$-ct-b constructs were significantly suppressed by 100 µM mepyramine (Fig. 2k–n). Together, these findings suggest that agonist and mechanical stimulation promote distinct conformational changes that correspond to distance increases between the two fluorophores.

**Mechanical $H_1R$ activation is independent of agonist binding.** To separate the response to histamine from the mechanosensitivity of the $H_1R$, we exchanged key amino acids (D116A and F433A) in the histamine-binding pocket[48]. The mutant FRET constructs were expressed at the plasma membrane (Supplementary Fig. 2) and exhibited indistinguishable BAL-induced FRET reductions and FRET efficiencies (Supplementary Fig. 3). The histamine-binding mutants retained mechanosensitivity, but did not respond to histamine, as evidenced by calcium imaging

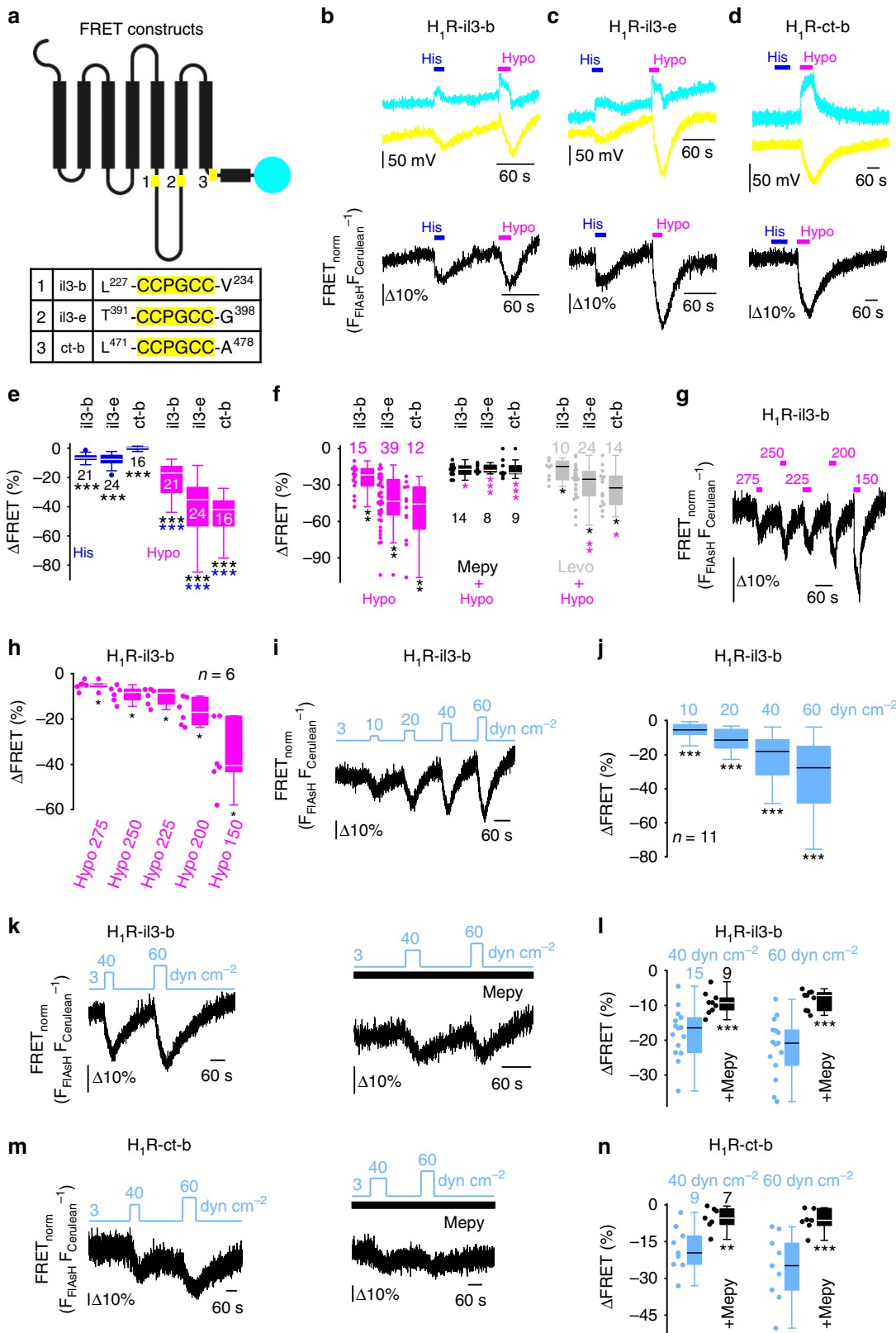

(Fig. 3a) and by electrophysiological whole-cell measurements[49] with HEK293 cells co-expressing the receptor-operated transient receptor potential classical cation channel 6 (TRPC6) as a readout system (Fig. 3b). Wild-type H$_1$Rs responded to histamine with prominent TRPC6 current increases (Fig. 3c). Mechanically and TRPC6 channel activator 1-oleoyl−2-acetyl-sn-glycerol

(OAG)–induced current responses did not differ between H$_1$R wild-type and H$_1$R mutant expressing cells. FRET measurements confirmed the loss of agonist sensitivity, while mechanosensitivity was preserved (Fig. 3d–g). There were no differences between mechanically induced FRET decreases of H$_1$R wild-type FRET constructs and the corresponding FRET constructs with disrupted

**Fig. 2 H₁R adopts distinct mechanically induced conformations. a** Schematic depiction of H₁R-FRET constructs with C-terminally attached cerulean and insertion of a FlAsH-binding motif ('CCPGCC') at different positions in the receptor: at the N-terminal beginning (il3-b) and at the C-terminal end (il3-e) of the third intracellular loop and at the beginning of the C-terminus (ct-b). Positions of the FlAsH-binding motif are highlighted in yellow. **b–d** Representative traces of histamine (His) and membrane stretch-induced (Hypo) fluorescence changes. Fluorescence intensity was measured as voltage of the transimpedance amplifier. Cyan traces represent cerulean and yellow traces FlAsH fluorescence. Black traces show normalized FRET signals. **e**, **f** Summaries of FRET signal changes. **e** Histamine-(blue) and hypoosmotically induced FRET changes (magenta). ***P < 0.001, blue asterisks; Wilcoxon signed-rank test to compare histamine- and mechanically induced FRET changes, and ***P < 0.001, black asterisks; Kruskal–Wallis test to compare histamine- and mechanically induced FRET changes. **f** Mechanically induced FRET changes in the absence (Hypo, magenta) and presence (Hypo + Mepy, black and Hypo + Levo, gray) of 30 μM mepyramine or 10 μM levocetirizine. *P < 0.05, **P < 0.01, ***P < 0.001, magenta asterisks; Mann–Whitney U test to compare hypoosmotically induced FRET responses in the presence and absence of inverse agonists and **P < 0.01, black asterisks; Kruskal–Wallis test to compare mechanically induced FRET changes. **g** Representative trace of hypoosmotically induced FRET signal changes of the H₁R-il3-b construct and application of different hypoosmotic solutions (275, 250, 225, 200 and 150 mOsm kg⁻¹). **h** Summary of FRET changes. *P < 0.05; Wilcoxon signed-rank test. **i**, **k**, **m** Representative FRET measurements with application of shear stress. **j**, **l**, **n** Summaries of shear stress-induced FRET signal changes in the presence and absence of 100 μM mepyramine (Mepy) (**j**) **P < 0.01; Wilcoxon signed-rank test. **l** and **n** **P < 0.01, ***P < 0.001; Mann–Whitney U test. Numbers indicate numbers of measured cells from at least 3 experimental days. n indicates the number of measured cells from three experimental days. Data are displayed as boxplots (median plus interquartile range (IQR) and whisker (max. 1.5-fold IQR)). See also Supplementary Figs. 2, 3, 4. Source data are provided as a Source Data file.

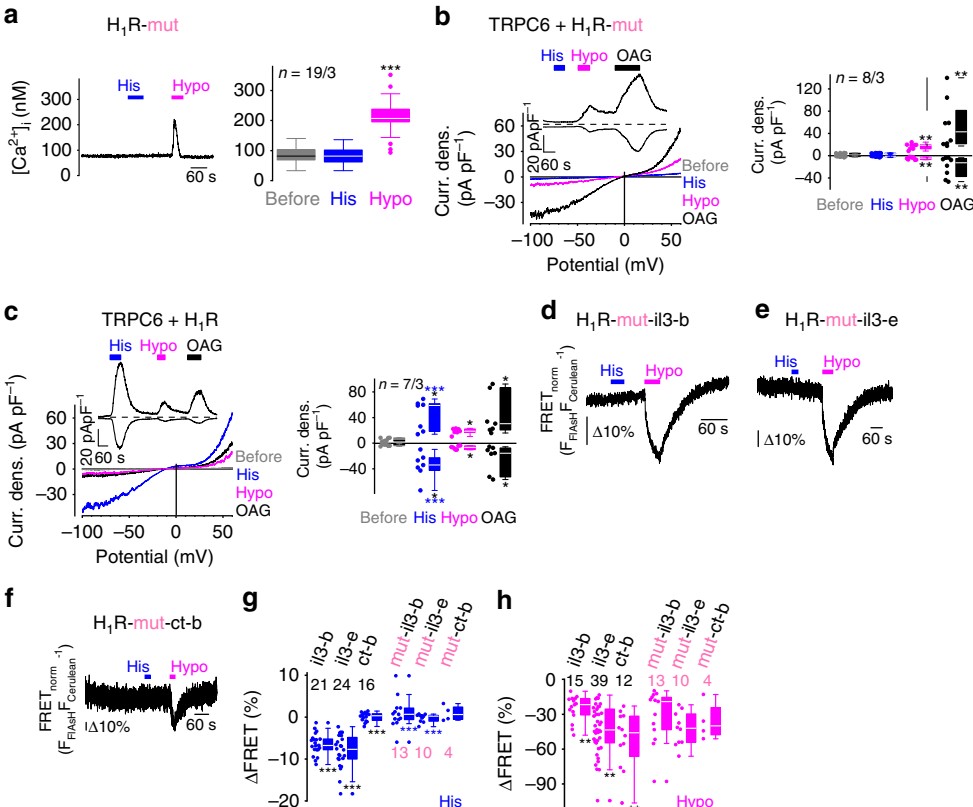

**Fig. 3 Mechanical H₁R activation is independent of agonist binding. a** Calcium imaging of fura-2 loaded HEK293 cells overexpressing H₁R with two amino acid exchanges (D116A and F433A; H₁R-mut) resulting in disruption of histamine binding. Representative trace of [Ca²⁺]ᵢ with applications of 100 μM histamine (His) and of hypoosmotic solution (Hypo, 150 mOsm kg⁻¹) and summary of [Ca²⁺]ᵢ are displayed. ***P < 0.001; Wilcoxon matched-pairs signed-rank test to compare to basal [Ca²⁺]ᵢ ('before'). **b**, **c** Whole-cell measurements of HEK293 cells co-expressing TRPC6 and the H₁R-mutant (**b**) or wild-type H₁R (**c**). Representative current density (Curr. dens.)-voltage curves (left) and current density-time courses (insets) with application of 100 μM histamine (His), hypoosmotic bath solution with 250 mOsm kg⁻¹ (Hypo) and of the TRPC6 activator 1-Oleoyl-2-acetyl-glycerol (OAG, 100 μM). Summaries of Current densities at holding potentials of ±60 mV before and during application of histamine, hypoosmotic bath solution and OAG (right). *P < 0.05, **P < 0.01, black asterisks; Wilcoxon matched-pairs signed-rank test to compare to basal current densities ('before') and ***P < 0.001, blue asterisks; Mann–Whitney U test to compare wild-type H₁R and H₁R-mutant. **d–f** Representative FRET measurements of indicated FRET constructs with impaired histamine binding. **g**, **h** Summaries of FRET signal changes induced by application of histamine (**g**) and of hypoosmotic solution (**h**). ***P < 0.001, blue asterisks; Mann–Whitney U test compared to wild-type and H₁R-mutant FRET constructs and **P < 0.01, ***P < 0.001, black asterisks; Kruskal–Wallis test to compare all wild-type and H₁R-mutant FRET constructs. **a–c** n = x/y indicates the sample size, where x is the number of measured cells and y is the number of coverslips from at least 3 experimental days. **a–c**, **g**, **h** Data are displayed as boxplots (median plus interquartile range (IQR) and whisker (max. 1.5-fold IQR)). See also Supplementary Figs. 2, 3. Source data are provided as a Source Data file.

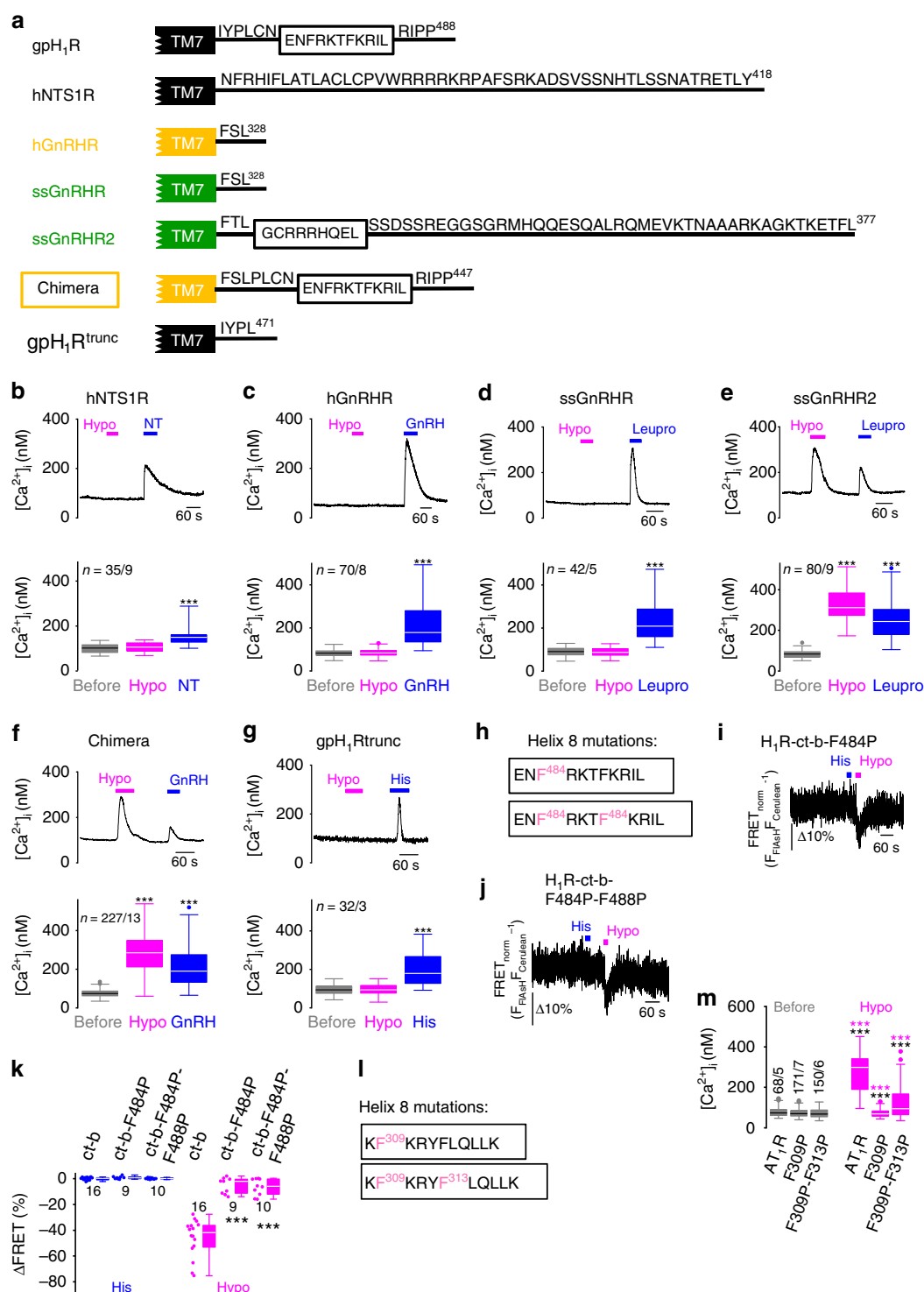

histamine-binding sites (Fig. 3h). These findings strongly indicate that mechanically induced $H_1R$ activation is agonist-independent as previously reported for $AT_1R$[15,17] and $CysLT_1R$[20].

**H8 is the essential structural motif for mechanosensing.** To identify motifs important for mechanosensation, we compared mechanosensitive and -insensitive GPCRs. Interestingly, the $G_{q/11}$-protein coupled human GPCRs NTS1R and GnRHR were insensitive to mechanical stimulation, but responded to agonists (Fig. 4a–c). These receptors differ from mechanosensitive GPCRs

like $H_1$, $AT_1$, $M_5$, $ET_A$ or $CysLT_1$ receptors[8,9] in that they lack the C-terminal H8. The role of H8 in mechanosensation has not been investigated, yet. However, an involvement of H8 in protein/lipid interaction, receptor internalization, dimerization or coupling to G-proteins was discussed (summarized in[50]). Notably, there are two different porcine GnRHR isoforms, a short (ssGnRHR) and a C-terminally extended version (ssGnRHR2), which either lack or contain a H8, respectively. While the short ssGnRHR was not mechanosensitive (Fig. 4d), the H8-containing ssGnRHR2 was (Fig. 4e). The GnRH superagonist leuprolide was administered as a positive control.

**Fig. 4 H8 is essential for mechanosensation of G$_{q/11}$-protein coupled receptors. a** C-terminal amino acid sequences of indicated GPCRs. gpH$_1$R, guinea pig H$_1$R; hNTS1R, human neurotensin 1 receptor; hGnRHR, human gonadotropin-releasing hormone receptor; ssGnRHR and ssGnRHR2, short and long isoforms of the swine GnRH receptor; chimera, hGnRHR fused to the C-terminus of gpH$_1$R; gpH$_1$R$^{trunc}$, C-terminally truncated gpH$_1$R. TM7; transmembrane domain 7. Black frames indicate helix 8 (H8). For ssGnRHR2, the H8 position was predicted using the YASPIN secondary structure prediction program. For gpH$_1$R and hNTS1R the H8 positions were predicted based on the crystal structures of the human H$_1$R and the rat NTS1R. Numbers indicate positions of last amino acids. **b–g** Calcium imaging with HEK293 cells overexpressing indicated receptors. Representative traces of [Ca$^{2+}$]$_i$. Applications of hypoosmotic solution (Hypo), of 200 nM neurotensin (NT, **b**), 200 nM GnRH (**c** and **F**), 200 nM leuprolide (Leupro, **d** and **e**) and 100 μM histamine (His, **g**) (upper panels). Summaries of [Ca$^{2+}$]$_i$ (lower panels). ***$P < 0.001$; Wilcoxon matched-pairs signed-rank test to compare to basal [Ca$^{2+}$]$_i$. **h** Amino acid sequence of H8 of the H$_1$R-c-tb construct with amino acid exchanges to proline. **i–k** FRET measurements of H8 mutant constructs. **l, j** Representative FRET measurements. **k** Summary of FRET signal changes. ***$P < 0.001$; Mann–Whitney $U$ test to compare to the H$_1$R-ct-b construct. **l** Amino acid sequence of H8 of the AT$_1$R with amino acid exchanges to proline. **m** Summary of [Ca$^{2+}$]$_i$ from HEK293 cells overexpressing AT$_1$R and H8 mutants. ***$P < 0.001$; black asterisks, Wilcoxon matched-pairs signed-rank test to compare to basal [Ca$^{2+}$]$_i$ before hypoosmotic stimulation. ***$P < 0.001$; magenta asterisks, Mann–Whitney $U$ test to compare hypoosmotically induced [Ca$^{2+}$]$_i$ to AT$_1$R WT. **b–g, m** n = x/y indicates the sample size, where x is the number of measured cells and y is the number of coverslips from at least 3 experimental days. **k, m** Numbers indicate the number of measured cells from at least 3 experimental days. **b–g, k, m** Data are displayed as boxplots (median plus interquartile range (IQR) and whisker (max. 1.5-fold IQR)). See also Supplementary Figs. 3, 5, 6. Source data are provided as a Source Data file.

We next generated a receptor chimera by fusing the H8-containing C-terminus of the mechanosensitive H$_1$R[51] to the human GnRHR (Fig. 4a). The receptor chimera was functional and responded with calcium transients to both mechanical and GnRH challenge (Fig. 4f). Conversely, removal of H8 from the H$_1$R by C-terminal truncation resulted in insensitivity to mechanical stimulation (Fig. 4g). These findings suggest that H8 is necessary for mechanosensation. To further test this hypothesis, we disrupted the α-helical structure of H8 in the H$_1$R-ct-b FRET construct by exchanging phenylalanines for prolines (F484P and F484P-F488P) (Fig. 4h). Both H8 mutants were functional and showed plasma membrane localization and unaltered FRET efficiencies (Supplementary Fig. 5 and Supplementary Fig. 3). However, the mutations resulted in significantly blunted mechanically induced FRET signals (Fig. 4i–k) and calcium transients (Supplementary Fig. 6a-d). To test whether H8 is essential for mechanosensitivity of AT$_1$Rs as well, we disrupted the structural integrity of H8 by select amino acid exchanges from phenylalanines to prolines (F309P and F309P-F3013P) (Fig. 4l, m). Performing calcium imaging, we observed that mechanically induced calcium increases were significantly reduced in AT$_1$R mutants while angiotensin II-induced calcium transients remained unaffected (Supplementary Fig. 6e-k). Thus, we conclude that H8 of AT$_1$R expressed in the vasculature of small resistance arteries is the essential structural motif for sensing mechanical forces and for converting them into vascular responses thereby contributing to the autoregulation of vessel tone[15,16,18,20]. Together, these findings suggest that mechanosensation of G$_{q/11}$-protein coupled receptors depends on the presence of a functional H8.

To analyze whether H8 is critical for mechanosensation of G$_{i/o}$- and G$_s$-protein coupled receptors as well, we next investigated the G$_{i/o}$-protein coupled human D$_2$R characterized by a C-terminal H8[52]. We performed whole-cell measurements on CHO-K1 cells that endogenously express low levels of potassium channels compared to HEK293 cells and over-expressed G$_{βγ}$-protein-regulated inward-rectifier potassium channels (Kir3.1/Kir3.2) alone or in combination with D$_2$R. Potassium current increases were used as a readout system. We noted that D$_2$R-expressing cells were mechanosensitive and showed prominent potassium current increases upon mechanical and agonist stimulation independent of the order of stimuli applied (Fig. 5a, b). The selective and potent D$_2$R blocker haloperidol significantly suppressed basal as well as hypoosmotically and dopamine-induced potassium currents (Fig. 5c). The initial dopamine stimulation was used as an expression control. Next, we analyzed CHO-K1 cells co-expressing Kir3.1/Kir3.2 and CXCR4. Crystal

structure analysis showed that human CXCR4 lacks a complete H8 and only possesses a single α-helical loop[53]. Interestingly, CXCR4-expressing cells were not mechanosensitive (Fig. 5d). Mechanical stimulation even reduced basal potassium currents of CXCR4-expressing (Fig. 5d) and of control cells expressing Kir3.1/Kir3.2 channels alone (Fig. 5e), consistent with a documented inhibitory effect of membrane stretch on Kir channel currents[54].

We also investigated G$_s$-protein coupled adenosine A$_{2A}$ receptor (A$_{2A}$R) receptors that possess a H8[55]. We employed an A$_{2A}$R-FRET construct with a FlAsH-binding motif at the C-terminal end of the third intracellular loop[56]. Mechanical stimulation caused FRET signal decreases that were larger than agonist-induced FRET signals (Fig. 5f). Blockade of the A$_{2A}$R by the potent and selective A$_{2A}$R blocker ZM 241385 (10 μM) significantly reduced mechanically induced FRET signals (Fig. 5g, j). Likewise, disruption of the α-helical structure of H8 by the exchange of threonine for a helix-breaking proline (T298P) (Fig. 5h) significantly reduced mechanically induced FRET signals (Fig. 5i, j), while agonist-dependent FRET signals were unaffected. Both A$_{2A}$R-FRET constructs showed unaltered maximal BAL-induced FRET decreases and FRET efficiencies (Supplementary Fig. 3). These findings support the model that H8 is a unifying structural requirement for mechanosensation of GPCRs.

## Discussion

This study considerably revises our current understanding of the role of GPCRs in mechanosensation, both at the physiological and the molecular level. At the physiological level, we demonstrate that H$_1$R is a sensor of fluid shear stress in the endothelium that contributes to flow-induced vasodilation. We used pharmacological tools as well as knock-out (H$_1$R$^{-/-}$ and H$_{1/2/3/4}$R$^{-/-}$) and smooth muscle-specific inducible knock-down (SmG$_{q/11}$$^{-/-}$) mice to show that shear stress activates mechanosensitive H$_1$R in endothelial cells in an agonist-independent manner, leading to G$_{q/11}$-protein activation, an increase in [Ca$^{2+}$]$_i$, NO production and vasodilation. To corroborate the suppression of flow-induced vasodilation by mepyramine applied at maximally effective concentrations, we additionally administered the more selective inverse agonist desloratadine which elicited comparable effects on flow-induced vasodilation. To rule out any off-target effects of the inverse agonists, we additionally analyzed H$_1$R$^{-/-}$ and H$_{1/2/3/4}$R$^{-/-}$ mice confirming the results obtained with the pharmacological tools.

Our findings suggest that the mechanosensitive H$_1$R acts as an endothelial mechanosensor that contributes to the autoregulation

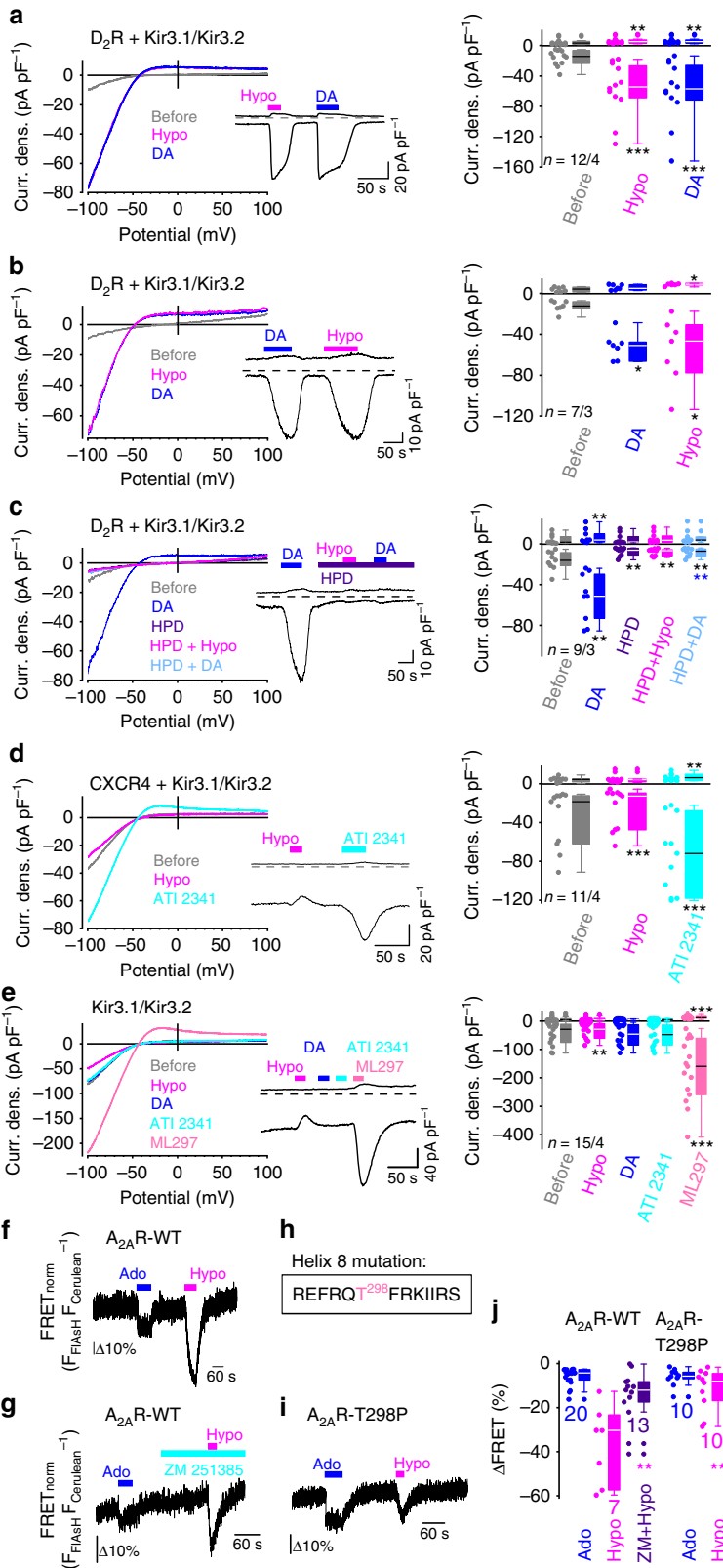

of blood vessels similar to the GPR68[19]. Other mechanosensitive GPCRs like sphingosine 1-phosphate and apelin receptors expressed in the endothelium were recently reported to be involved in vascular development[12] and endothelial cell polarization[10,11]. However, not all GPCRs act as primary mechanosensors in the endothelium. Recent findings suggest that the mechanosensitive ion channel PIEZO1 expressed in endothelial cells acts as a sensor of shear stress involved in ATP- and adrenomedullin-mediated vasodilation[37,38]. In this case, GPCRs might play a role as mechanotransducers in the endothelium. However, it was even reported that PIEZO1 might be activated downstream of a mechanosensitive GPCR complex thus serving as

**Fig. 5 H8 is essential for mechanosensation of $G_{i/o}$- and $G_s$-protein coupled receptors. a–e** Whole-cell measurements of CHO-K1 cells overexpressing Kir3.1/Kir3.2 channels alone (**e**) or in combination with the $D_2R$ (**a–c**) or the CXCR4 (**d**). (left) Representative current density (Curr. dens.)-voltage curves and current density-time courses (insets) with application of hypoosmotic bath solution with 250 mOsm $kg^{-1}$ (Hypo), 10 µM dopamine (DA), 10 µM haloperidol (HPD), 10 µM ATI 2341 and 3 µM of the Kir channel activator ML297 (above). (right) Summaries of current densities at holding potentials of ± 100 mV before and during mechanical and agonist stimulation. $^*P < 0.05$, $^{**}P < 0.01$, $^{***}P < 0.001$; Wilcoxon matched-pairs signed-rank test to compare to basal current densities before stimulation and $^{**}P < 0.01$, blue asterisks to compare dopamine effects in the presence or absence of haloperidol. **f, g, i, j** FRET measurements of the $A_{2A}R$ FRET construct ($A_{2A}R$-WT, **f** and **g**) and of the $A_{2A}R$-T298P FRET construct ($A_{2A}R$-T298P, **i**). **h** Position of amino acid exchange in H8 of the $A_{2A}R$ FRET construct to disrupt the helix structure. **f, g, i** Representative FRET measurements with application of 10 µM adenosine (Ado) and of hypoosmotic solution (Hypo) and of 10 µM of the $A_{2A}R$ blocker ZM 241385 (ZM). **j** Summary of adenosine and of mechanically induced FRET signal changes. $^{**}P < 0.01$, magenta asterisks; Mann–Whitney $U$-test to compare mechanically induced FRET changes. **a–e, j** Data are displayed as boxplots (median plus interquartile range (IQR) and whisker (max. 1.5-fold IQR)). **a–e** $n = x/y$ indicates the sample size, where $x$ is the number of measured cells and $y$ is the number of coverslips from at least three experimental days. **j** Numbers indicate the number of measured cells from at least three experimental days. See also Supplementary Fig. 3. Source data are provided as a Source Data file.

a mechanotransducer[57]. Not only GPCRs and ion channels, but several other proteins might be involved in mechanosensation of endothelial cells[36]. Since flow-induced vasodilation is of the utmost importance for regulation of vessel perfusion, it is likely that different mechanosensitive signaling pathways have evolved and that several mechanosensors cooperate to sense blood flow and to convert the mechanical force into cellular signals[36]. Our previous findings suggested that intrinsically mechanosensitive GPCRs in the vasculature contribute to myogenic vasoconstriction by about 64%[8,20]. Hence, about 36% of myogenic tone is mediated by other mechanosensitive signaling pathways. Moreover, different mechanosensitive signaling pathways might have developed in different vascular beds. For example in podocytes, GPCRs are neither involved in mechanosensation nor -transduction[58].

To elucidate whether $H_1Rs$ are involved in mechanosensation of endothelial cells, we first employed HUVEC since these cells are commonly accepted as being sensitive to fluid shear stress. Under physiological conditions veins and venules are exposed to shear stress in the range of 1–29 dyn $cm^{-2}$ [59,60]. Conduit arteries usually experience shear stress up to 10 dyn $cm^{-2}$ while arterioles are exposed to shear stress up to 70 dyn $cm^{-2}$[60]. Thus, shear stress of 4 and 20 dyn $cm^{-2}$ we applied on HUVEC (see Fig. 1b, c) was in the physiological range. However, the first switch from no flow to flow conditions by application of shear stress of 4 dyn $cm^{-2}$ on HUVEC should be regarded as non-physiological because under physiological conditions endothelial cells do not experience such changes from no flow to flow. Interestingly, the more physiological increase of shear stress from 4 to 20 dyn $cm^{-2}$ provoked additional calcium responses. The intraluminal shear stress of 4.8 an 8.8 dyn $cm^{-2}$ applied on mesenteric artery segments corresponded to the physiological range for conduit arteries. Performing FRET measurements by analyzing $H_1R$-FRET constructs, we applied shear stress starting from constant basal levels of 3 dyn $cm^{-2}$. Shear stress was then increased in a stepwise fashion up to 60 dyn $cm^{-2}$ reflecting the physiological range of arterioles. $H_1R$-FRET constructs were sensitive to shear stress and to membrane stretch and there was a correlation between the strength of the stimulus and the resultant FRET signal, thus meeting a critical criterion for intrinsically mechanosensitive proteins[9]. Our previous studies showed that hypoosmotically induced membrane stretch and direct membrane stretch[15,58] elicit comparable effects. Therefore, we used hypoosmotically induced membrane stretch as a fast screening method to elucidate the molecular principles of mechanosensation of GPCRs. This approach has its limitations, because in the vasculature, a rapid decrease of tonicity cannot be regarded as a physiological stimulus and the supplemented sugar alcohol mannitol in the isosmotic bath solution might exert additional antioxidative effects. Thus, as a proof-of-principle, we applied shear stress as a more physiological distinct mechanical stimulus.

Interestingly, the inverse agonist mepyramine at maximal concentrations did not fully abolish shear stress-, but hypoosmotically induced calcium increases in HUVECs. Since laminar and oscillatory shear stress can cause distinct cellular signals (summarized in[36,61]), one may speculate that different mechanical stimuli induce distinct mechanosensitive signaling pathways. Potentially, membrane stretch, which is not the major mechanical stimulus for endothelial cells, mainly causes activation of $H_1R$, while shear stress additionally activates other mechanosensors apart from $H_1R$ resulting in calcium increases. Obviously, regulation of mechanosignaling in endothelial cells is complex and involves different signaling pathways. The mechanisms by which distinct mechanical stimuli can activate GPCRs are still elusive. However, membrane stretch and shear stress may both influence the lipid order of the plasma membrane[62,63]. It has been reported that shear stress may increase membrane fluidity[62] and that membrane stretch may change the lateral pressure profile of the plasma membrane[64]. These alterations of the phospholipid bilayer may allow integral membrane proteins like GPCRs to adopt active conformations. Further studies are required to unravel these mechanisms in greater detail.

$H_1R$ antagonists or inverse agonists are widely used as antihistaminic drugs. First generation antihistaminic drugs might cause dizziness and orthostatic hypotension due to their anti-α-adrenergic side effects. However, for the $H_1R$ selective second generation antihistaminic drugs it is not documented that they significantly influence arterial blood pressure. Our findings suggest that antihistaminic drugs prevent flow-induced vasodilation but do not initiate active vasoconstriction. Since regulation of vessel tone is vitally important, several mechanisms act in concert to keep blood pressure levels constant and the signaling pathways involved may vary between different vascular beds. Thus, dysregulation of the blood pressure under antihistaminic medication might be masked. Interestingly, it was reported that the intake of antihistaminic drugs prior to sporting activity prevents post-exercise hypotension[65] and reduces blood flow[66] indicating that under certain conditions impaired flow-induced vasodilation by antihistaminic drugs might indeed influence blood pressure. Mechanosensitive GPCRs like $AT_1Rs$ are involved in several physiological and pathophysiological states such as myogenic vasoconstriction[15,16,18,20] and load-induced cardiac hypertrophy[17] or preeclampsia[23]. In line with our mechanistic concept, $AT_1R$ also require an intact H8 for mechanosensation. A better understanding of the mechanisms underlying their mechanosensitivity is of utmost importance and might help to further improve medication.

At the molecular level, we provide strong evidence in favor of H8 as the essential structural feature of mechanosensitive GPCRs. Notably, we observed the largest mechanically induced FRET signal decreases when analyzing the $H_1R$-c-tb construct with the

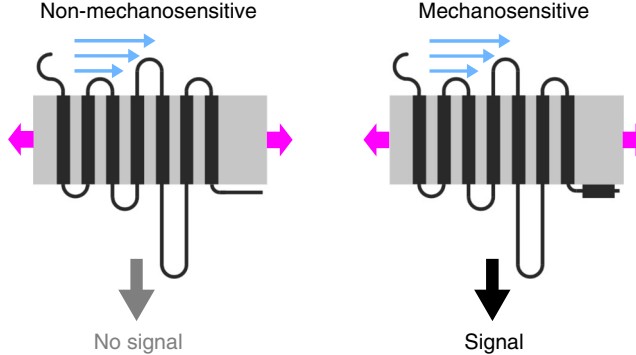

**Fig. 6 Model of mechanically induced GPCR activation.**
Mechanosensation of GPCRs depends on the presence of an intact H8. GPCRs that lack H8 are non-mechanosensitive (left). An intact H8 stabilizes the mechanosensitive receptor conformation resulting in activation of GPCRs (right). Mechanical forces like membrane stretch and shear stress might cause elongation of H8 leading to G-protein activation and subsequent signal transduction resulting in large signals. GPCR is depicted in black. H8 is highlighted as black rectangular box. Gray box indicates the plasma membrane. Blue arrows display shear stress and magenta arrows display membrane stretch.

FlAsH-binding site located just proximal to H8. This observation is consistent with the hypothesis that mechanical stimulation entails an elongation of H8. Such an assumption is compatible with the notion that α-helices may act as constant force springs that can unfold and refold progressively as the length increases or decreases[67]. However, at present, FRET signal changes by rotation of the fluorophore cannot be ruled out. Further studies are necessary to elucidate the mechanism of H8 elongation in more detail. An analysis on the downstream effects of H8 elongation would be important to gain deeper insight into the pathomechanisms of mechanically induced diseases.

In summary, we show that the presence of an intact H8 is a pivotal prerequisite for mechanosensation of GPCRs (Fig. 6). H8 is critically involved in mechanosensation of $G_{q/11}$-, $G_{i/o}$- and $G_s$-protein coupled receptors. GPCRs lacking a functional H8 due to impairment of the structural integrity of H8 or by C-terminal truncation are not mechanosensitive and, conversely, transfer of H8 to a formerly non-mechanosensitive GPCR confers mechanosensitivity. Our findings indicate that mechanical forces induce distinct conformational changes in mechanosensitive GPCRs which might result in H8 elongation and subsequent G-protein activation and signal transduction resulting in differential cellular responses (Fig. 6).

## Methods

**Materials.** Poly-L-lysine hydrobromide(P1524), gelatin type B (G9391), histamine dihydrochloride (H7250), human gonadotropin-releasing hormone (GnRH, L7134), leuprolide (L0399), adenosine (A9251), ATI 2341 (SML1720), ML297 (SML0836), dopamine hydrochloride (H8502), neurotensin actetate (N6383), levocetirizine dihydrochloride (L7795), dimercaprol (2,3-Dimercapto-1-propanol, BAL, 64046), fura-2-acetoxymethyl ester (fura-2, 47899), 1-Oleoyl-2-acetyl-sn-glycerol (OAG, 47989), desloratadine (D1069) and angiotensin II (A-9525) were purchased from Sigma-Aldrich. U46619 (538944) was purchased from Calbiochem. YM254890 (AG-CN2–0509-M001) was purchased from AdipoGen Life Sciences, mepyramine maleate (0660). ZM 241385 (1036) and haloperidol hydrochloride (0931) were purchased from Tocris. FlAsH was a gift from Carsten Hoffmann and additionally purchased from Cayman Chemicals.

**Mice.** All experiments involving animals at the Ludwig Maximilians University Munich were performed in accordance with the EU Animal Welfare Act and were approved by the governmental oversight authority Regierungspräsidium Oberbayern on animal care. Tamoxifen-inductions of tamoxifen-inducible, smooth muscle-specific $G_{q/11}$-protein knock-down mice[45] were approved by the Regierungspräsidium Darmstadt (permit no. B2–1069). Adult (10–15-week old) male

mice were kept under standard light/dark cycle with food and water ad libitum. All mice were maintained in IVC housing and manipulated in class IIA laminar flow biosafety cabinets to prevent pathogen cross contamination. The mice were regularly tested for pathogens by use of sentinel animals, which were screened using either serology or PCR on tissue, feces and pelt swabs. Representative animals were also used for health surveillance. The following mice were used: wild-type (C57BL/6 J, Janvier), tamoxifen-inducible, smooth muscle-specific $G_{q/11}$-protein knock-down mice (tamoxifen-treated SMMHC-CreER$^{T2}$;$Gnaq^{flox/flox}$;$Gna11^{-/-}$)[45], Histamine H1 receptor gene-deficient mice ($Hrh1^{tm1Wtn}$; $H_1R^{-/-}$)[43] and Histamine $H_1$, $H_2$, $H_3$, $H_4$ receptor quadruple gene-deficient mice ($H_{1/2/3/4}R^{-/-}$) which were generated from Saligrama et al. by intercrossing B6.129P-$Hrh1^{tm1Wat}$, B6.129P-$Hrh2^{tm1Wat}$, B6.129 P2-$Hrh3^{tm1Twl}$ and B6.129P-$Hrh4^{tm1Thr}$ mice[44]. The number of animals used is a minimum necessary to provide adequate data to test the hypothesis of this project. The number of mice were minimized according to the animal welfare committee wherever possible. At least three mice per genotype and experimental group were used for statistical purposes.

**Myography.** After decapitation of anesthetized male mice, mesentery was removed and transferred to cold (4 °C), oxygenated (with carbogen: 95% $O_2$ and 5% $CO_2$), physiological salt solution (PSS) with the following composition (in mM): 119 NaCl, 25 NaHCO$_3$, 4.7 KCl, 1.2 KH$_2$PO$_4$, 1.2 MgSO$_4$, 1.6 CaCl$_2$ and 11.1 glucose (pH 7.4 after purging by carbogen). Mesenteric arteries of first and second order branches devoid of surrounding fat were cannulated on extended soda lime glass pipettes (∅ 1.2 mm, 1413021, Hilgenberg) filled with PSS allowing application of intravascular pressure and flow. Cannulated vessels were mounted onto a pressure myograph (DMT 111 P, DMT) and outer vessel diameters were determined performing video microscopy using MyoView software (DMT). Measurements were performed with continuous superfusion at 37 °C. Viability of the vessels was tested by applying 60 mM potassium chloride (KCl) solution with the following composition (in mM): 63.7 NaCl, 25 NaHCO$_3$, 60 KCl, 1.2 KH$_2$PO$_4$, 1.2 MgSO$_4$, 1.6 CaCl$_2$ and 11.1 glucose (pH 7.4 after purging by carbogen). Only arteries showing 60 mM KCl-induced constriction of at least 30% were further investigated. Vessels were pre-constricted at intravascular pressure of 50 mmHg under no flow conditions (inflow and outflow pressures were 50 mmHg) either with the thromboxane A2 receptor agonist U46619 (20 nM) in PSS or with 35 mM KCl solution (in mM: 88.7 NaCl, 25 NaHCO$_3$, 35 KCl, 1.2 KH$_2$PO$_4$, 1.2 MgSO$_4$, 1.6 CaCl$_2$ and 11.1 glucose (pH 7.4 after purging by carbogen)) which both resulted in approximately 20% vasoconstriction. After 30 min, intravascular flow was stepwise increased and the outer vessel diameters were monitored. Flow was increased to Δ5 mmHg (inflow pressure was 53 mmHg and outflow pressure was 48 mmHg) for 10 min and to Δ15 mmHg (inflow pressure was 58 mmHg and outflow pressure was 43 mmHg) for 10 min and decreased to no flow conditions (inflow and outflow pressures were 50 mmHg) for 5 min. Subsequently, the presence of functional endothelium was verified by 300 µM acetylcholine-induced vasodilation in pre-constricted arteries. Flow rates were determined by collecting the intravascular perfusate at Δ5 mmHg and Δ15 mmHg which corresponded to 27.1 ± 3.9 µl min$^{-1}$ (mean ± sem) or 32.4 ± 5.9 µl min$^{-1}$ (mean ± sem). Shear stress was calculated using the inner diameters of artery segments according to Koop et al.[68] using the following equation:

$$\tau = \frac{32 * Q * \eta}{d^3 * \pi}, \tag{1}$$

where $\tau$ is the shear stress, $Q$ is the flow in m$^3$ s$^{-1}$, $\eta$ is the viscosity of $1 \times 10^{-3}$ N s m$^{-2}$ and $d$ is the internal diameter of the vessel. The calculated values in N m$^{-2}$ were converted into dyn cm$^{-2}$. This resulted in shear stress values of 4.8 ± 0.5 dyn cm$^{-2}$ (mean ± sem) at Δ5 mmHg and 8.8 ± 1.1 dyn cm$^{-2}$ (mean ± sem) at Δ15 mmHg. Finally, vessels were incubated with calcium free PSS solution with 3 mM EDTA at no flow conditions serving as a reference to determine maximal passive dilative diameters at intraluminal pressure of 50 and 120 mmHg. The maximal passive dilative diameters at 50 mmHg were used for normalization. All blockers or inhibitors were intravascularly applied and incubated under no flow conditions for 30 min prior to measurements. The mean outer vessel diameter at no flow conditions was 275 ± 6 µm (mean ± sem) ($n = 75$).

**Histamine enzyme immunoassay.** To determine total intravascular histamine concentration in vessel perfusates, a commercially available enzyme immunoassay with a detection limit of 1.8 nM was performed following the manufacturer's instructions (Histamine ELISA Kit, ARG80457, Arigo Biolaboratories). All samples ($n = 5$ after pre-constriction with 20 nM U46619 and $n = 6$ after pre-constriction with 35 mM KCl) were analyzed in triplets. Histamine was not detectable in all perfusates.

**Determination of nitrate concentration.** To monitor total intravascular nitrate concentrations in vessel perfusates from wild-type arteries derived from C57BL/6 J mice, wild-type arteries treated with mepyramine or levocetirizine or $H_1R^{-/-}$ arteries as a measure of NO production, a commercially available nitrate/nitrite fluorometric assay kit (Cayman Chemical, 780051) was used according to the manufacturer's instructions. The nitrate concentration was normalized to the total volume of perfusates collected during flow-induced vasodilation after pre-constriction with U46619. All samples were analyzed in triplets.

**Cell culture and transfections.** Human embryonic kidney (HEK293) cells (from Leibniz-Institute DSMZ, T293, DMSZ no. ACC 635) were maintained in Earl's minimal essential medium (Sigma-Aldrich) and Chinese hamster ovary (CHO-K1) cells (from Leibnitz-Institute DSMZ, DMSZ no. ACC 110) were maintained in Ham's F-12 medium (Sigma-Aldrich) with 100 U ml$^{-1}$ penicillin and 100 µg ml$^{-1}$ streptomycin supplemented with 10% fetal calf serum (FCS, Gibco) and 2 mM glutamine with 100 U ml$^{-1}$ penicillin and 100 µg ml$^{-1}$ streptomycin supplemented with 10% fetal calf serum (FCS, Gibco) and 2 mM glutamine. Human umbilical vein endothelial cells (HUVEC) (Promocell, C-12250) were cultivated according to the supplier's instructions with Endothelial Cell Growth Medium (Promocell) containing 2% FCS. Endothelial cells were used for calcium imaging experiments at passage 2–6. HUVEC were seeded onto poly-L-lysine (PLL, Sigma-Aldrich)-coated glass coverslips (Ø 24 mm, neoLab) or into gelatin-coated flow chambers (µ-slide I 0.4, ibidi). All cells were held at 37 °C in a humified atmosphere with 5% CO$_2$. For FRET measurements stable monoclonal HEK293 cell lines expressing the following constructs were used: C-terminally cerulean-tagged guinea pig histamine H$_1$ receptor (gpH$_1$R, accession no S68706) with binding motif for FlAsH at the beginning (H$_1$R-il3-b) and at the end of the third intracellular loop (H$_1$R-il3-e) and at the begin of the C-terminus (H$_1$R-ct-b). Moreover, monoclonal cell lines were used with C-terminally cerulean-tagged H$_1$Rs with disrupted histamine-binding site by amino acid exchanges from aspartate at position 116 and phenylalanine at position 433 to alanine (D116A and F433A) and with binding motif for FlAsH at the beginning (H$_1$R-mut-il3-b), at the end of the third intracellular loop (H$_1$R-mut-il3-e) and at a proximal position of the C-terminus (H$_1$R-mut-ct-b). The monoclonal HEK293 (from Leibniz-Institute DSMZ, 293, DSMZ no. ACC 305) cell lines were cultured in supplemented Dulbecco's minimal essential medium (Sigma-Aldrich) additionally containing 800 µg ml$^{-1}$ G418 (Invitrogen). For FRET measurements, cells were seeded onto PLL-coated coverslips (Ø 30 mm, neoLab) 24–48 h prior to experiments. Either monoclonal cell lines were used or HEK293 cells that were transiently transfected using lipofection with GeneJuice® (Merck Millipore) according to the manufacturer's instructions. Cells were transfected with 1 µg cDNA coding for the FRET constructs: H$_1$R-ct-b-F484P, H$_1$R-ct-b-F484P-F488P, human adenosine A$_{2A}$ receptor with binding motif for FlAsH at the end of the third intracellular loop (il3-e) that was C-terminally fused to cerulean (A$_{2A}$R-WT)[56], A$_{2A}$R-FRET construct with one amino acid exchange in helix 8 to disrupt the helix structure (A$_{2A}$R-T298P), human angiotensin II AT$_1$R (accession no.: NM_000685) C-terminally fused to mVenus (AT$_1$R), AT$_1$R with amino acid exchanges to disrupt H8 (AT$_1$R-F309P and AT$_1$R-F309P-F313P). For calcium imaging experiments either monoclonal cell lines were used or HEK293 cells were transiently transfected at confluency of about 70–80% 24–48 h prior to the measurements. Cells were seeded onto PLL-coated coverslips (Ø 24 mm, neoLab) 24 to 48 h prior to experiments and transfected with 1 µg cDNA coding for one of the following receptors: gpH$_1$R, gpH$_1$R with amino acid exchanges (D116A-F433A, 'H$_1$R-mut'), human neurotensin 1 receptor (hNTS1R, accession no.: AY429106), human gonadotropine-releasing hormone (GnRH) receptor (hGnRHR, accession no.: NM_000406), swine GnRH receptor (ssGnRHR, accession no.: NM_214273), swine GnRH receptor isoform 2 (ssGNRHR2, accession no.: JQ828975), human GnRHR chimera fused to the C-terminus of the gpH$_1$R (hGnRHR-gpH$_1$R$^{C-terminus}$), gpH$_1$R with truncated C-terminus (gpH$_1$R$^{trunc}$), with gpH$_1$R-ct-b-construct or gpH$_1$R-ct-b-constructs with amino acid exchanges in helix 8 to disrupt the helix structure (H$_1$R-ct-b-F484P and H$_1$R-ct-b-F484P-F488P), with AT$_1$R with amino acid exchanges to disrupt H8 (AT$_1$R-F309P and AT$_1$R-F309P-F313P). If the receptor construct was not tagged to cerulean, 0.1 µg pEYFP-N1 (Clontech) was co-transfected. In case of H$_1$R-ct-b constructs and of the truncated H$_1$R, 0.5 µg G$_{qα}$-protein[69], which was a gift from Bruce Conklin, was co-transfected. For electrophysiological whole-cell measurements, HEK293 cells were co-transfected with 2 µg human TRPC6 (accession no.: NP_004612) in pIRES2-EGFP expression vector (Clontech) and 0.5 µg H$_1$R-mut or CHO-K1 cells were co-transfected with 1 µg rat inward-rectifier potassium channel Kir3.1 (accession no.: NM_031610) and human inward-rectifier potassium channel Kir3.2 (accession no.: NP_002231) in bicistronic pIRES expression vector (Clontech) (Kir3.1-IRES-Kir3.2, 'Kir3.1/3.2') and with 0.5 µg human chemokine type 4 receptor (hCXCR4, accession no.: NM_001295) or human dopamine D2 receptor (hD$_2$R, (accession no.: NM_000795) and 0.2 µg cerulean in pcDNA3.1 expression vector (Clontech).

**Molecular biology.** Site-directed mutagenesis was performed on the guinea pig histamine H$_1$ receptor in pcDNA3.1(+) (accession number: BAA03669), on the human GnRH receptor in pcDNA3.1 (accession number: L07949), on the human adenosine A$_{2A}$R FRET construct[56] and on the human AT$_1$R in pcDNA3.1 (accession number: NM_000685) using the QuikChange system (Stratagene). For FRET constructs, the cDNA encoding for mCerulean 1 was fused to the C-terminus of the H$_1$R receptor by PCR of the cDNA. For this, the H$_1$R was amplified by PCR using the following primers: 5'-ATT AAG CGG CCG CAT GTC TTT CCT CCC AGG AAT-3' sense and 5'-TAA TCT CGA GAG GGG GGA TAC GCA GGA TCC-3' antisense. mCerulean 1 was also amplified by PCR using the following primers: 5'AAA TCT CGA GGT GAG CAA GGG CGA GGA GCT-3' sense and 5'-GGC CCT CTA GAT TAC TTG TAC AGC TCG TCC A-3' antisense. Both inserts were subcloned into the vector pcDNA3.1(+) (Invitrogen). To insert the FlAsH-binding motif 'CCPGCC' into the H$_1$R cDNA either the cutting sites PmlI

(for H$_1$R-il3-e) or BsiWI (for H$_1$R-il3-b) were used or a SacII cutting site was generated by site-directed mutagenesis (for ct-b) which allowed insertion of the FlAsH-binding motif. The primers used for site-directed mutagenesis to generate a new restriction site for creation of the ct-b constructs are listed in the Supplementary Methods section. hAT$_1$R-F309P was used as a template to perform a second amino acid exchange resulting in the double mutant hAT$_1$R-F309P-F313P. gpH$_1$R-D116A construct was used as a template for second amino acid exchange resulting in the double mutant gpH$_1$R-D119A-F433A. The FlAsH-binding motif was inserted in the third intracellular loop of the receptor at two positions after leucine L227 (H$_1$R-il3-b construct) or after threonine T391 (H$_1$R-il3-e construct) and at the beginning of the C-terminus after leucine L471 by subcloning. The following oligonucleotides were annealed and subcloned into the gpH$_1$R-cerulean construct: 5'-GTA CTA TGT TGT CCG GGG TGT TGT-3' sense and 5'-GTA CAC AAC ACC CCG GAC AAC A-3' antisense for the il3-b construct, 5'-GTG TTG TCC GGG GTG TTG TGG-3' sense and 5'-CCA CAA CAC CCC GGA CAA C-3' antisense for the il3-e construct, and 5'-TTT GTT GTC CGG GGT GTT GTG CGC-3' sense and 5'-GCA CAA CAC CCC GGA CAA CAA AGC-3' antisense for the ct-b construct. All receptor cDNAs were subcloned into mammalian expression vector pcDNA3.1(+). The following amino acid exchanges were introduced by site-directed mutagenesis: from aspartate at position 116 (D116A) and from phenylalanine at position 433 to alanine (F433A) in guinea pig H$_1$R to disrupt agonist binding site of the receptor[48], from phenylalanine to proline (F484P and F484P-F488P) in guinea pig H$_1$R, from threonine to proline in the A$_{2A}$R-FRET construct (A$_{2A}$R-T298P) and from phenylalanine to proline (F309P and F309P-F313P) in human AT$_1$R to disrupt helix 8 structure. For calcium imaging experiments a chimera of the hGnRHR and the C-terminus of the gpH$_1$R was built. For this, the complete C-terminus of gpH$_1$R was amplified by PCR using the following primers: 5'- ATT AAG AAT TCA ATG AGA ATT TCA GGA AGA CCT TCA AGA GGA TCC TGC GTA TCC CCC CTC TCG AGA ATT A-3' sense and 5'-TAA TTC TCG AGA GGG GGG ATA CGC AGG ATC CTC TTG AAG GTC TTC CTG AAA TTC TCA TTG AAT TCT TAA T-3' antisense and fused to the C-terminus of hGnRHR using XhoI and EcoRI cutting sites. After subcloning the missing stop codon was generated performing site-directed mutagenesis. Moreover, gpH$_1$R was C-terminally truncated by insertion of a stop codon by site-directed mutagenesis after the first four C-terminal amino acids. All primer sequences for site-directed mutagenesis are listed in the Supplementary Methods section (Supplementary Information Table 1). All cDNA constructs used in the present work were confirmed by sequencing.

**Calcium imaging.** Intracellular free calcium concentrations were determined in fura-2-acetoxymethyl ester (fura-2, 5 µM; Sigma-Aldrich) loaded HEK293 cells and HUVECs. Glass coverslips (Ø 24 mm) were mounted on the stage of a monochromator-equipped (Polychrome V, TILL-Photonics) inverted microscope (Olympus IX 71 with an UPlanSApo 20 × /0.85 oil immersion objective). Fluorescence was recorded with a 14-bit EMCCD camera (iXON 885 K, Andor). Fura-2 fluorescence was excited at 340 and 380 nm. Free intracellular calcium concentrations [Ca$^{2+}$]$_i$ were calculated according to the following formula from Grynkiewicz et al.[70]:

$$c = \beta * Kd * \frac{(R - R_{min})}{(R_{max} - R)}, \qquad (2)$$

where $c$ is the [Ca$^{2+}$]$_I$, $\beta$ is the ratio of the maximal and minimal fluorescence values at 380 nm, $Kd$ is the equilibrium dissociation constant at room temperature (264 nM), $R$ is the ratio of the fluorescence values at 340 and 380 nm, $R_{min}$ is the minimal value of the fluorescence ratios at 360 and 380 nm determined in calcium-free solution in the presence of 5 µM ionomycin and $R_{max}$ is the maximal value of the fluorescence ratios at 360 and 380 nm determined in high calcium solution containing 95 mM CaCl$_2$. Cells were continuously superfused at room temperature with HEPES-buffered saline (HBS) solution containing (in mM) 140 NaCl, 5 KCl, 1 MgCl$_2$, 2 CaCl$_2$, 5 glucose, 10 HEPES (pH 7.4 with NaOH). For measurements with hypoosmotically induced membrane stretch isosmotic and hypoosmotic bath solutions were used: Isosmotic solution (Iso) contained (in mM) NaCl 55, KCl 5, CaCl$_2$ 2, HEPES 10, glucose 10, MgCl$_2$ 1 (pH 7.4 with NaOH) which was supplemented with mannitol to 300 mOsm kg$^{-1}$. The hypoosmotic solution (Hypo) had the same salt concentration without added mannitol resulting in an osmolality of 149–152 mOsm kg$^{-1}$. For agonist stimulation 100 µM histamine, 200 nM neurotensin, 200 mM GnRH, 200 nM leuprolide or 100 nM angiotensin II were applied. Shear stress measurements were performed with HUVEC, which were seeded into gelatin-coated flow chambers (µ-slide I 0.4, ibidi). Cells were superfused with HBS solution. Measurements started with no flow conditions. Using a peristaltic-pump (ISM827B, Ismatec) perfusion rates of 3.0 and of 15.2 ml min$^{-1}$ were applied that corresponded to 4 dyn cm$^{-2}$ and 20 dyn cm$^{-2}$.

**Quantitative real-time PCR (qPCR).** Total RNA from HUVEC was isolated using the Tri Reagent (Sigma-Aldrich). First strand synthesis was carried out with random hexamers as primers, using REVERTAID reverse transcriptase (Fermentas). The primers pairs used for the amplification of specific fragments from the first strand synthesis are provided in the Supplementary Methods section (Supplementary Information Table 2). Primers for human H$_1$R, H$_2$R, H$_3$R, H$_4$R, M$_1$R, M$_3$R, M$_5$R, ET$_A$R, ET$_B$R, V$_{1A}$R, V$_2$R, B$_1$R, B$_2$R, P2Y$_1$R, CysLT$_1$R, CysLT$_2$R, UT$_2$R, alpha$_{2A}$R,

beta$_2$R, beta$_3$R, GPR68, AT$_1$R, PTH$_1$R, D$_5$R, and for the endothelial cell markers platelet endothelial cell adhesion molecule (CD31), von Willebrand factor (vWF) and vascular endothelial cadherin (cadherin 5) and three references hypoxanthin phosphoribosyltransferase 1 (Hprt1), tyrosine 3-monooxygenase/tryptophan 5-monooxygenase activation protein, zeta polypeptide (Ywhaz), and succinate dehydrogenase complex, subunit A (Sdha) were used. Real-time polymerase chain reaction (RT-PCR) was performed using the master mix from the Absolute QPCR SYBR Green Mix kit (ABgene). Ten picomole of each primer pair and 0.2 μl of the first strand synthesis was added to the reaction mixture, and PCR was carried out in a light-cycler apparatus (Light-Cycler 480, Roche) using the following conditions: 15 min initial activation and 45 cycles of 12 s at 94 °C, 30 s at 50 °C, 30 s at 72 °C, and 10 s at 80 °C each. Fluorescence intensities were recorded after the extension step at 80 °C after each cycle to exclude fluorescence of primer dimers melting lower than 80 °C. All primers were tested by using diluted complementary DNA (cDNA) from the first strand synthesis (10–1000 ×) to confirm linearity of the reaction and to determine particular efficiencies. Data were calculated as percentage of the geometric mean expression of the three references. PCR products were verified by agarose gel electrophoresis. Crossing points were determined by the software program. All experiments were performed in triplets. We performed independent experiments with HUVEC preparations from maximal nine different single donors.

**FlAsH labeling**. The labeling was done according to Hoffmann et al.[56]. For this, HEK293 cells transiently or stably expressing FRET constructs were washed twice with phenol red-free HBS solution (HBSS, Sigma-Aldrich) and incubated at 37 °C and 5% CO$_2$ for 1 h with 2 ml 250–500 nM FlAsH in HBSS supplemented with 12.5 μM 1,2-ethane dithiol (EDT) in HBSS. After incubation, cells were washed once with HBSS. To reduce the background signal, cells were subsequently incubated at 37 °C and 5% CO$_2$ for 10 min with 2 ml 250 μM EDT in HBSS to reduce unspecific labeling. Subsequently, cells were washed twice with HBSS and FRET measurements were immediately performed.

**Förster resonance energy transfer (FRET)**. Dynamic intramolecular FRET measurements in intact cells were performed according to Hoffmann et al.[56]. FRET measurements using FlAsH as an acceptor and cerulean as a donor were performed with cells that were previously seeded onto PLL-coated glass coverslips (Ø 30 mm) or into PLL-coated flow chambers (μ-slide VI 0.1, cat. no. 80666, ibidi). Flow chambers were continuously superfused with HBS solution using a syringe pump (Perfusor secura FT, B. Braun). Basal flow rate was 1.9 ml h$^{-1}$, which corresponded to 3 dyn cm$^{-2}$. Increasing flow rates of 5.4, 11.4, 22.2 and 33.6 ml h$^{-1}$ corresponded to 10, 20, 40 and 60 dyn cm$^{-2}$. Glass coverslips were transferred into a measuring chamber that was filled with isosmotic bath solution (Iso). Cells on coverslips were continuously superfused with Iso. Hypoosmotic bath solution with an osmolality of 150 mOsm kg$^{-1}$ and agonists were applied using a rapid pressure driven (0.7 bar) focal superperfusion device ALA-VM8 (ALA Scientific Instruments) which allowed solution exchange within 5–10 ms. Flow chamber and measuring chamber were mounted on the stage of an inverted microscope (IX 70, Olympus) with an UPlanSApo 100 ×/1.40 oil immersion objective. Upon excitation at 430 nm with Polychrome V (TillPhotonics) fluorescence intensities at 480 ± 20 and 535 ± 15 nm were measured with a dual-emission photometry system (Till Photonics). Emission was measured as voltage of the transimpedance amplifier from the photodiodes and was collected by EPC10 amplifier (HEKA) with the Patchmaster software (HEKA). FRET ratios were measured as the ratio of the corrected FlAsH and cerulean emissions. Normalized FRET (FRET$_{norm}$ with F$_{FlAsH}$ F$_{Cerulean}$$^{-1}$) were calculated as the corrected emission intensities at 535 ± 15 and 480 ± 20 nm (beam splitter DCLP 505 nm) on excitation at 430 nm (beam splitter DCLP 460 nm). Excitation time was 4.6 ms and sampling rate was up to 20 Hz. Fluorescence traces were corrected for bleaching. Bleed through of cerulean into the 535 nm channel and of FlAsH into the 480 nm channel and crosstalk were subtracted to yield a corrected FRET ratio. Ratio changes were further analyzed and regarded as FRET signals if the course of the single fluorescence traces developed in opposite directions. Moreover, only cells showing membrane staining of cerulean fluorescence were selected for further analysis. To obtain maximal FRET efficiencies of the FlAsH-labeled FRET constructs 10 mM 2,3-dimercapto-1-propanol (Dimercaprol, British-Anti-Lewisite, BAL) was applied during FRET measurements which caused stripping of FlAsH from the tetracysteine binding motif. BAL completely dequenched the cerulean fluorescence which allowed to estimate the FRET efficiency[56]. FRET efficiencies (E) were calculated according to the following formula with 'FCerulean, before' as the bleaching-corrected fluorescence intensity of cerulean before application of BAL and 'FCerulean, BAL' as maximal fluorescence intensity of cerulean after application of BAL:

$$E = 1 - \frac{F\text{Cerulean, before}}{F\text{Cerulean, BAL}}. \tag{3}$$

**Electrophysiological whole-cell measurements**. Conventional whole-cell patch-clamp recordings were carried out at room temperature (23 °C). Transfected HEK293 cells were superfused with an isosmotic bath solution containing 110 mM NaCl, 5 mM CsCl, 1 mM MgCl$_2$, 2 mM CaCl$_2$, 10 mM glucose, 10 mM HEPES (pH 7.4 with NaOH) supplemented with mannitol to 300 mOsm kg$^{-1}$. For hypoosmotically induced membrane stretch a hypoosmotic solution without added

mannitol was used which resulted in an osmolality of 249–253 mOsm kg$^{-1}$. The pipette solution for TRPC6 expressing HEK293 cells contained 120 mM CsCl, 9.4 mM NaCl, 0.2 mM Na$_3$-GTP, 1 mM MgCl$_2$, 3.949 mM CaCl$_2$, 10 mM BAPTA (100 nM free Ca$^{2+}$) and 10 mM HEPES (pH 7.2 with CsOH), resulting in an osmolality of 295 mOsm kg$^{-1}$. Transfected CHO-K1 cells were superfused with the following isosmotic bath solution containing 25 mM KCl, 1 mM MgCl$_2$, 10 mM HEPES, 10 mM glucose, 0.1 mM CaCl$_2$ and 88 mM NaCl (pH 7.4 with NaOH) supplemented with mannitol to 300 mOsm kg$^{-1}$. The hypoosmotic solution had the same salt concentrations but contained no mannitol which resulted an osmolality of 247–252 mOsm kg$^{-1}$. The pipette solution for Kir3.1/Kir3.2 expressing CHO-K1 cells contained 80 mM KCl, 50 mM K-glutamate, 3.949 mM CaCl$_2$, 2 mM MgCl$_2$, 10 mM HEPES, 10 mM BAPTA (100 nM free Ca$^{2+}$) and 2 mM ATP magnesium salt (pH 7.2 with KOH), resulting in an osmolality of 293 mOsm kg$^{-1}$. Liquid junction potentials were +4.0 mV and +7.3 mV for TRPC6 and Kir channel recordings, respectively and were corrected by the Patchmaster software. Data were collected with an EPC10 patch-clamp amplifier (HEKA) using the Patchmaster software (HEKA). Current density-voltage relations were obtained from triangular voltage ramps from −100 to +60 mV with a slope of 0.4 V s$^{-1}$ applied at a frequency of 1 Hz in case of TRPC6 channel recordings. In case of Kir channel recordings current voltage relationships were obtained from voltage ramps from −100 to 100 mV with a slope of 0.5 V s$^{-1}$ at a frequency of 2 Hz. All data were acquired at a frequency of 5 kHz after filtering at 1.67 kHz. The current density-voltage curves and the current density amplitudes at ±60 mV or ±100 mV were extracted at minimal or maximal currents, respectively. Patch pipettes made of borosilicate glass (Science Products) had resistances of 2.0–3.5 MΩ.

**Confocal microscopy**. Confocal microscopy was performed on a Leica TCS SP5 II system (Leica, Germany). PLL-coated glass bottom dishes (μ-slide 4 or 8 well, ibidi) with transiently and stably transfected HEK293 cells were treated with ice-cold 100% methanol and cells were fixed for 5 min at −20 °C. Cells were washed once after fixation and kept in PBS and immediately after this, confocal images were taken with a 63 × 1.40 UV oil immersion objective. Probes were excited with 405 nm (6% laser intensity) while emission was measured at 521 to 557 nm.

**Statistical analysis**. Calculations and statistical analysis were performed using the OriginPro 7.5 and OriginPro 2019b (OriginLab) and the GraphPad Prism 7 and Prism 8 software (GraphPad). No statistical methods were used to predetermine sample size. Outliers were identified and removed using iterative Grubb's test. If not otherwise stated, data are displayed as boxplot with median plus interquartile ranges (percentiles 25 and 75%). Whiskers display the maximal 1.5-fold interquartile range. Data points higher than the 1.5-fold interquartile range are displayed as individual data points over whiskers. For kinetic analysis, the changes of FRET signals during agonist and hypoosmotic stimulation were fitted with a mono-exponential function applying simplex algorithms and Levenberg-Marquardt iterations, until no reduction of Chi-square was notable. All data were tested for normal distribution using Shapiro-Wilk test. If data was not normally distributed, differences between more than two groups were assessed by Kruskal–Wallis test. Either Wilcoxon matched-pairs signed-rank test or Mann–Whitney U test as unpaired post hoc test were used to estimate differences between two groups. A P-value less than 0.05 was considered significant for all analyses. In case of calcium imaging measurements, significant differences were assumed relevant only if differences between medians were ≤4 nM.

**Reporting summary**. Further information on research design is available in the Nature Research Reporting Summary linked to this article.

## Data availability

Data supporting the findings of this manuscript are available from the corresponding authors upon reasonable request. A reporting summary for this Article is available as a Supplementary Information file. The source data underlying Figs. 1a, 1a, 1e-j, 2e, 2f, 2h, 2j, 2l, 2n, 3a-a, 3g, 3h, 4b-g, 4k, 4m, 5a-e, 5j and Supplementary Figs. 1c-f, 2a-d, 2l, 3k-n, 4b-e, 5a, 5b, 6d and 6k are provided as a Source Data file.

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

## Acknowledgements

The authors thank Erhard Wischmeyer for providing Kir3 channel cDNA and Julie Straub for subcloning Kir3.1 and Kir3.2 into the pIRES vector. We thank Cory Teuscher for allocating $H_{1/2/3/4}R^{-/-}$ mice, and we thank Takeshi Watanabe and Perti Panula for providing $H_1R^{-/-}$ mice. The authors also thank Diana Gabriel and Robert Mayer for their contribution to calcium and FRET measurements. This work was supported by the German Research Foundation (Deutsche Forschungsgemeinschaft) (TRR-152, TRR-166 project C02 and project no. 406028471) and by the German Centre for Cardiovascular Research (DZHK B 16–007 Extern).

## Author contributions

M.M.yS., U.S., and S.E. designed and performed experiments and analyzed the data. U.S., M.M.yS., and T.G. wrote the manuscript. T.G., U.S., and M.M.yS. supervised the study. U.S. performed electrophysiological measurements. S.E., J.B., M.W., L.D., and U.S. performed calcium imaging. S.E., M.W., J.B and M.M.yS. performed mutagenesis and S.E., J.B., and L.D. performed FRET measurements. L.D. performed measurements of flow-induced vasodilation and quantitative RT-PCRs. S.E., U.S., and L.D. performed NO and Histamine assays. C.H., N.Z., S.E., and M.M.yS. performed and analyzed FRET kinetic measurements. A.W. and S.O. provided $G_{q/11}$ knock-down mice.

## Competing interests

The authors declare no competing interests.
