## [Peer Review File · Nature Communications]

Reviewers' Comments:

Reviewer #1:

Remarks to the Author:

The authors display a substantial body of work on a difficult and controversial topic. The concept they propose is not novel – indeed it is relatively old - but the big effort to provide new important information in this area should be appreciated. Nevertheless, there are several concerns that I advise the authors to carefully consider.

The Introduction (and manuscript generally) is polarized and narrow in its coverage of prior work in this field, failing to mention major efforts by numerous investigators who have committed much of their careers to understanding the topic of shear stress and force sensing in mammalian biology. Aside from the problem of not recognising and valuing this other work, it is a concern that the approach could mislead and confuse readers, and that it might even mislead the authors themselves.

It is important to consider how the proposed hypothesis relates to the prominent recent work published by co-author Dr Offermanns, which suggested that shear stress responses of this type are instead mediated by PIEZO channel detecting force that is then signalled via ATP-release and P2Y receptor activation. There is substantial potential for confusing the field without proper consideration of this matter. The authors should attempt a rationalisation of their ideas.

In this study there is extensive use of hypotonic shock as a surrogate for shear stress (in some cases extreme hypotonic shock). This is a concern because it is probably a poor surrogate that does not mimic shear stress and which has other effects mediated by changes in tonicity and not mechanical force. Consideration should be given to omitting all of these tonicity data unless the authors choose to be much open about studying mechanisms of cellular response to tonicity rather than shear stress.

Shear stress used in the FRET studies is high, and higher than that used for the endothelial cell experiments. In Figure 2 there appears to have been effect of shear stress at 10 dyne/cm² but not 3 dyne/cm² but the most convincing effects are for higher shear stresses. The strong drift in the baseline is also a concern in these recordings. The effects shown may not be relevant to the physiological shear stress experienced by endothelial cells.

The FRET studies are potentially interesting and the technical challenges are appreciated but the authors risk creating a false impression by extensive dependence on their over-expression system and cell lines exposed to quite extreme stimuli. It is recommended to be more cautious in style and conclusions.

The authors should consider and discuss the possibility of an alternative hypothesis that H8 affects responses indirectly by modifying the local PIP₂ concentration. This risk is most for the over-expression studies but might also be relevant to the native system where receptor like histamine receptors are strongly expressed, as emphasised by the authors.

The final conclusion (in the Abstract) about H8 having therapeutic potential is premature and not reflective of the primary effort of the study.

The endothelial cells used in this study are referred to as "primary" but this is misleading because the cells were commercially-obtained HUVECs passaged 2 to 6 times.

The suggestion that 0.1 mM mepyramine is selective requires justification and should not simply be assumed. It could have modulated other candidate shear stress sensor mechanisms?

For Figure 1 contraction studies the data need to be presented more clearly without

normalisation/subtraction and showing data for other properties of the arteries to convince the reader that the recordings were of suitable quality and that non-specific effects were avoided. Sections of original tracings would be nice to see as raw data examples.

Figure S1: the traces are mistakenly labelled with micro M instead of micro m. Dilation is evident before the shear stress was applied, suggesting inaccurate labelling or lack of correlation with shear stress. Original traces are low quality with poor resolution, giving concern about the reliability of the original data.

In various places there are statements without supporting references or data. For example on page 4 "Histamine was not detectable in the vessel perfusates using a commercially available enzyme immunoassay with a detection limit 35 of 1.8 nM suggesting that H1R is activated agonist-independently by shear stress leading to Gq/11-protein activation and NO production thus resulting in flow-induced vasodilation."

In the Results text for FRET studies it should be made more obvious at the outset which types of cells were used.

-Page 6 typo: ", but. Notably ..."

Overall, a much more concise presentation of data and text would help the authors convince the reader of the main strengths of their case. Consideration should be given to focus on a smaller number of the most important findings.

Reviewer #2:

Remarks to the Author:

The authors have performed a study showing that the C-terminal helix 8 (H8) provided mechanosensitivity to the histamine H1 GPCR. This makes this GPCR a new are endothelial sensor of shear stress in endothelial cells. The H8 might also be the link between the mechanosensitive GPCRs and the production of vasodilators by the endothelium.

The study is done properly using appropriate approaches. The data support the conclusion and the discussion is clear enough with reasonable arguments.

Nevertheless, some concerns remain.

1 – Figure 1: the authors should also show that mepyramine or desloratadine reduce flow-mediated nitrate production. Furthermore, it is also important to show the possible effect (or absence of effect) of the KO (and of mepyramine or desloratadine) of the contractility of the arteries and on the vasodilation induced by agonists (acetylcholine for example).

2 – Similarly, fig 1A; it would interesting to see the expression level of the other mechanosensitive GPCRs that you cite in the discussion (the AT1R for example)

3 – Discussion: link between cholesterol and mechanosensitivity: might more complexe in hypercholesterolemic patients. If I follow you rhyptohesis, their endothelium should be more sensitive to shear stress and then it should be better protected (shear increases eNOS level...etc).

4 – Do you expect an equivalent H8 in AT1R or the other mechanosensitive GPCRs with the same role and pathway?

5 – AT1R is expected to play a role in pressure-induced tone more than in flow-mediated

relaxation. As both shear stress and hypo-osmotic shock activate the H8-associated pathway, do the authors expect a role for H8 in myogenic tone?

Reviewer #3:
Remarks to the Author:

Review report Erdogmus et al.

In the present MS the authors present compelling evidence on the role of in the endothelial H1R receptor as a sensor of shear stress, mediating flow-induced vasodilation. Moreover, the mechanosensitivity can be measured as well in transfected HEK-293 cells. Using sophisticated single cell FRET measurements using conformational H1R biosensors, the authors show that the mechanosensitive response induces a different H1R conformation compared to the GPCR agonist histamine. Moreover, the authors present compelling evidence on the role of helix 8 in the mechanosensitive properties of the H1R and a number of other GPCRs and have tried to resolve the actual mechanism involved. Overall, a very comprehensive study with state-of-the-art techniques that is revealing an very interesting feature of GPCR proteins.

Major comments

Abstract: in the last sentence the authors suggest that “H8 might display a therapeutic target”. I feel that that there is no actual foundation for this in the present work. There is no evidence that a small part of a GPCR can be seen as a specific target. Do the authors perhaps mean that it might constitute an interesting new region of the GPCR proteins, that could potentially be targeted as well? However, there is no evidence that this can be done without affect normal agonist function on the GPCR target or how one should try to get specificity for GPCRs with H8. I strongly advise to change the final sentence of the abstract (and also the final sentences in the discussion).

Fig. 1B-E: Mepyramine is inhibiting the calcium responses to histamine and hypotonic solutions completely, but not the calcium responses to shear stress (1B. lower panel). What to make of this? Should one argue that in HUVECs the shear stress response is inducing calcium responses via a different mechanism than the hypotonic solution? Can the authors comment on this and mention something about these differences in the text?

Fig. 2F: the authors show that the conformational changes of the H1R sensors after hypotonic solutions can only partially be inhibited by H1 blockers. As the H1R sensors still give good calcium responses (Fig.S2) upon histamine addition, it would be good to know if also a hypotonic solution is giving calcium responses and if that can be fully blocked by the used concentrations of mepyramine and levo-cetirizine.

Fig. 3B: next to the calcium measurements and the H1R sensors, the authors now introduce a third method (TRPC6 readout) to look at the functionality of the used H1R mutant. Since these mutants are already known to knock-out histamine functionality, I am not sure if this is very useful, especially as WT responses are not shown for comparison. I suggest to keep focus on the calcium tracings and the H1R sensors and omit the electrophysiology in this figure.

Page 6: The authors also investigate if Gi/o and Gs coupled receptors are mechanosensitive, if the GPCR also contains H8. One important aspect that is however not shown is if these mechanosensitive responses can be counteracted by D2 or A2A antagonists. This is a relatively

simple experiment that would strengthen the H1R case of blockade of mechanosensitive H1R signalling and shear-stress induced vasodilation by H1 antagonist.

Page 7: the part on the role of PIP2 is in my opinion the least developed part of the story. The part on wortmannin is questionable as at 20 uM, wortmannin will hit many targets (see ChemBI or PubChem), so the effect cannot be attributed to PIP2 levels. I would omit this part. Next, potential binding partners for PIP2 binding are mutated. For the R mutations, data on membrane expression is missing and should be shown. Finally, some effects of cholesterol are reported on the effects of hypotonic solution on one FRET sensor, but a characterization of the effects of H1R agonist responses on FRET sensors or effects on H1R-mediated calcium responses are not shown. This part feels relatively incomplete and could be omitted from the paper in my opinion as well.

Discussion, page 8. In the discussion the authors get back to the original findings on the vasodilation effect after shear stress. The authors try to translate their results with cholesterol to a clinical setting, but for the studied mechanosensitive GPCRs no attention is given to the H1R. In contrast, the authors mention the AT1R instead. Yet, H1 antagonists have been used by millions of people and data on cardiovascular effects or risk factors will undoubtedly be available. In view of the present focus of the MS, I suggest that the authors pay some attention to this point.

Minor comments

Page 6, line 1: *mechanicallyinduced* should be *mechanically-induced*

Page 6, line 11: incomplete sentence

Page 7, line 37: I do not think that the terminology of "gene-deficient mouse" is appropriate?

Point-to point response to the reviewers:

Reviewer #1

The authors display a substantial body of work on a difficult and controversial topic. The concept they propose is not novel – indeed it is relatively old - but the big effort to provide new important information in this area should be appreciated. Nevertheless, there are several concerns that I advise the authors to carefully consider.

Although mechanosensitivity of GPCRs is known for about 15 years until now, the mechanisms underlying the mechanosensitivity of GPCRs on the molecular level have remained largely elusive. In our study, we aimed to shed light on the molecular structures that are essentially involved in mechanosensation of GPCRs. For the first time, we could show that the C-terminal helix 8 is a critical structural motif essential for mechanosensitivity of GPCRs. We are convinced that the topic of our study is of utmost importance and may provide a solid foundation for a better understanding of the physiological and pathophysiological roles of mechanosensitive GPCRs.

We appreciate the recommendations of reviewer 1 and we have thoroughly revised our manuscript accordingly. In the whole manuscript, we changed the presentation of our data according to the Journals guidelines and we now display data as boxplot analysis plus interquartile range. We show individual data points if N number is ≤ 10 . Moreover, we applied Grubb's test to identify and remove outliers. Significances were tested again after removal of the outliers. Moreover, colors were changed from Red and Green to Magenta and Blue to support colorblind readers.

The Introduction (and manuscript generally) is polarized and narrow in its coverage of prior work in this field, failing to mention major efforts by numerous investigators who have committed much of their careers to understanding the topic of shear stress and force sensing in mammalian biology. Aside from the problem of not recognising and valuing this other work, it is a concern that the approach could mislead and confuse readers, and that it might even mislead the authors themselves.

We appreciate the concern raised by the reviewer. We have now expanded the introduction section to cover prior work on mechanosensation of endothelial cells (page 3 lines 24-33). However, we are confined to a maximal number of 70 references by the manuscript guidelines.

It is important to consider how the proposed hypothesis relates to the prominent recent work published by co-author Dr Offermanns, which suggested that shear stress responses of this type are instead mediated by PIEZO channel detecting force that is then signalled via ATP-release and P2Y receptor activation. There is substantial potential for confusing the field without proper consideration of this matter. The authors should attempt a rationalisation of their ideas.

Dr. Offermanns' work about PIEZO 1 channels as mechanosensors in the endothelium is indeed very important. However, we already mentioned and discussed his work in the discussion section. In the revised version of our manuscript, we mention his work not only in the discussion, but also in the introduction section by citing his recent publications concerning PIEZO1 and P2Y²¹ and adrenomedullin² receptors in endothelial cells. In these studies, PIEZO1 emerged as a mechanosensor and GPCRs act as mechanotransducers in the endothelium that are activated by mechanically induced agonist release. However, it was recently reported that PIEZO1 might be activated downstream of a mechanosensitive GPCR complex thus serving as a

mechanotransducer³. Since vasoregulation is vitally important, it is likely that different regulation mechanisms of vessel tone have evolved. Our previous findings suggest that intrinsically mechanosensitive GPCRs in the vasculature contribute to myogenic vasoconstriction by about 64%^{4, 5}. Therefore, the remainder of about 36% of myogenic tone must be mediated by other distinct mechanosensitive signaling pathways. Moreover, in different vascular beds different signaling pathways for mechanosensation have developed. For example, in podocytes GPCRs are not involved in mechanosensation or -transduction⁶. We conclude that several mechanosensors cooperate to sense blood flow and to convert mechanical forces into cellular signals. We tried to discuss the complexity of the topic in more detail and we have expanded the discussion section in this respect (page 10, lines 14ff).

The aim of our study was the elucidation of the mechanism of mechanosensitivity of GPCRs motivated by the discovery of a novel physiological role of mechanosensitive H₁R_s. We found that H₁R_s are sensitive to membrane stretch and to shear stress. Therefore, we started out to elucidate the principles underlying mechanosensation of H₁R_s in particular and of GPCRs in general. Our focus was not primarily on the dissection of the principles of shear stress. We performed a more general approach aiming to analyze the molecular mechanisms underlying mechanosensation of GPCRs. We tried to define our rationale more precisely in the introduction section (page 3, line 34ff and page 4, lines 1-2).

In this study there is extensive use of hypotonic shock as a surrogate for shear stress (in some cases extreme hypotonic shock). This is a concern because it is probably a poor surrogate that does not mimic shear stress and which has other effects mediated by changes in tonicity and not mechanical force. Consideration should be given to omitting all of these tonicity data unless the authors choose to be much open about studying mechanisms of cellular response to tonicity rather than shear stress.

We agree that hypoosmotically induced membrane stretch is not suitable as a surrogate for fluid shear stress. In previous studies, we found that hypoosmotically induced membrane stretch has the same effects as direct membrane stretch induced by application of positive pressure through the patch pipette^{6, 7} or by raising the patch pipette in the whole cell configuration⁵ exerting local membrane stretch. Since direct membrane stretch had the same effect as hypoosmotically induced membrane stretch, we used hypotonicity as a fast screening method. Here we show that two different mechanical stimuli - membrane stretch and shear stress –both activate H₁R_s. We applied shear stress as a more physiological, but distinct mechanical stimulus and used this stimulus as a proof-of-principle for mechanosensitivity of H₁R. Shear stress applied on HUVECs endogenously expressing H₁R_s as well as on HEK293 cells over-expressing H₁R-FRET constructs caused intracellular calcium increases or FRET signal decreases similar to hypoosmotically induced membrane stretch, suggesting that both stimuli stabilize active receptor conformations that promote G-protein activation and subsequent signaling.

Future more detailed and focused studies will have to elucidate the underlying mechanisms by which these distinct mechanical stimuli activate GPCRs. One may speculate that both mechanical stimuli influence the lipid order of the plasma membrane^{8, 9}. An increased membrane fluidity induced by shear stress⁸ and changes of the lateral pressure profile by membrane stretch¹⁰ might allow integral membrane proteins like GPCRs to adopt active receptor conformations. We have expanded the discussion section accordingly (page 11, lines 10-28).

Shear stress used in the FRET studies is high, and higher than that used for the endothelial cell experiments. In Figure 2 there appears to have been effect of shear stress at 10 dyne/cm² but not 3 dyne/cm² but the most convincing effects are for higher shear stresses. The strong drift in the baseline is also a concern in these recordings. The effects shown may not be relevant to the physiological shear stress experienced by endothelial cells.

Under physiological conditions, veins and venules are exposed to shear stress in the range of 1-29 dyn cm⁻²^{11, 12}. Conduit arteries usually experience shear stress up to 10 dyn cm⁻² while arterioles are exerted to shear stress up to 70 dyn cm⁻²¹². Performing FRET measurements by analyzing H₁R-FRET constructs, we applied shear stress starting from constant basal levels of 3 dyn cm⁻². We did not monitor the switch from no flow conditions to shear stress of 3 dyn cm⁻² which we regarded as non-physiological. Shear stress was then increased stepwise up to 60 dyn cm⁻² to mimic physiological changes of blood flow. These shear stress values are in the physiological range of arterioles. The observed correlation between the strength of the mechanical stimulus and the resultant FRET signal is an important criterion for intrinsically mechanosensitive membrane proteins¹³. These statements are included in the discussion section (page 10, line 331ff and page 11, lines 1-9).

The drift in some FRET signal baselines occasionally occurred at the end of long FRET measurements (≥ 5 minutes) and we believe that these shifts were due to focus drifts. We decided not to display these artefacts at the end of some FRET measurements and therefore, we have shortened the duration of the traces (see Figures 2B, 2I, 2K and Supplementary Figure 4A).

The FRET studies are potentially interesting and the technical challenges are appreciated but the authors risk creating a false impression by extensive dependence on their over-expression system and cell lines exposed to quite extreme stimuli. It is recommended to be more cautious in style and conclusions.

We appreciate the reviewer's advice and we tried to be more cautious in style and conclusions. As mentioned above, we used FRET constructs to elucidate the molecular mechanisms underlying mechanosensation of GPCRs. For this, we applied different mechanical stimuli (hypoosmotically induced membrane stretch and shear stress) that should serve as a proof-of-principle for mechanosensitivity.

The authors should consider and discuss the possibility of an alternative hypothesis that H8 affects responses indirectly by modifying the local PIP₂ concentration. This risk is most for the over-expression studies but might also be relevant to the native system where receptor like histamine receptors are strongly expressed, as emphasised by the authors.

We thank reviewer 1 for this suggestion. In agreement with the editor Dr. Mieck and because reviewer 4 gave us the advice to omit all data regarding the role of cholesterol and of PIP₂ for mechanosensation, we have deleted this data in the revised manuscript. We agree that the role of PIP₂ needs a more detailed analysis.

The final conclusion (in the Abstract) about H8 having therapeutic potential is premature and not reflective of the primary effort of the study.

We agree that the conclusion that H8 might have therapeutic potential is premature and we have deleted the sentence in the abstract.

The endothelial cells used in this study are referred to as "primary" but this is misleading because the cells were commercially obtained HUVECs passaged 2 to 6 times.

HUVEC cells are specified as primary endothelial cells on the homepage of the supplier Promocell (<https://www.promocell.com/product/human-umbilical-vein-endothelial-cells-huvec/>). In the Materials and Methods section, we stated that these cells were purchased from the company Promocell and that we used HUVECs cultivated and passaged up to 6 times. To avoid any misunderstandings, we now clearly specify the cells as HUVECs in the results section (page 5, line 5). We used HUVECs as a model to analyze the mechanosensitivity of endogenously expressed H₁R_s. Since these cells are of premature nature and not fully differentiated, we verified

our results in a more physiological setting analyzing flow-induced vasodilation of isolated murine mesenteric artery segments. These statements are now included in the results section (page 5, line 22f).

The suggestion that 0.1 mM mepyramine is selective requires justification and should not simply be assumed. It could have modulated other candidate shear stress sensor mechanisms?

We agree that 100 μ M mepyramine is a very high concentration. However, we additionally applied the inverse agonist desloratadine, which likewise reduced flow-induced vasodilation similar to mepyramine. However, since we cannot rule out off-target effects induced by the two inverse agonists, we subsequently analyzed H₁R and H_{1/2/3/4}R gene-deficient mice to sort out the role of endothelial H₁Rs for mechanosensation. Since these mice also showed impaired flow-induced vasodilation, we were able to corroborate the effects of the inverse agonists. We now discuss the use of mepyramine in more detail (page 10, lines 8-13).

For Figure 1 contraction studies the data need to be presented more clearly without normalisation/subtraction and showing data for other properties of the arteries to convince the reader that the recordings were of suitable quality and that non-specific effects were avoided. Sections of original tracings would be nice to see as raw data examples.

In Figure 1 we display the summaries of the vasodilation measurements for the sake of clarity. Showing all traces in one Figure (30 traces for Figure 1H and 45 traces for Figure 1I) would be confusing. However, we already show representative original raw data traces without normalization in the Supplementary Figures section (Supplementary Figure 1). The summaries displayed in Figure 1 are normalized using the maximal passive dilative diameters measured at 50 mmHg for normalization. We analyzed vessels with comparable outer diameters at no flow conditions of 275 ± 6 μ m (mean \pm sem). This information was stated in the Material and Methods section.

To demonstrate that the vessels used for experiments were viable and comparable, we now display the following additional vessel parameters (see Supplementary Figure 1C-1F):

1. Maximal vasoconstrictions induced by 60 mM potassium chloride. These vasoconstrictions were used as a control at the beginning of the measurements prior to application of any blockers. Only vessels showing constriction of at least 30% were regarded as viable and were selected for further analysis.
2. Acetylcholine-induced vasodilations determined after application of the shear stress protocol.
3. Maximal vasodilations induced by application of Ca²⁺-free solution at intraluminal pressures of 50 mmHg.
4. Maximal vasodilations induced by application of Ca²⁺-free solutions at intraluminal pressures of 120 mmHg.

No significant differences between wild-type vessels in the presence and absence of mepyramine or desloratadine and between different genotypes were observed indicating that all vessels were viable and comparable.

Figure S1: the traces are mistakenly labelled with micro M instead of micro m. Dilation is evident before the shear stress was applied, suggesting inaccurate labelling or lack of correlation with shear stress. Original traces are low quality with poor resolution, giving concern about the reliability of the original data.

We apologize for this mistake and we have changed the labeling accordingly. The example trace for H₁R gene-deficient arteries in Supplementary Figure 1B shows a small but reversible increase before first application of shear stress. We now display a different original example trace showing vasodilation of a H₁R gene-deficient artery. All original traces are raw data traces which are displayed with the full resolution of 1 μ m which is the maximal resolution given by the pressure

myograph (DMT 111P) from the company DMT. The resolution of 1 μm is now stated in the Figure legend to Supplementary Figure 1B and 1C. We decided to forgo smoothing our traces just as we did in previous publications^{7, 14} to prevent suggesting a non-existing precision. Several other groups using pressure myography likewise show representative non-smoothed original traces^{15, 16, 17, 18}.

In various places there are statements without supporting references or data. For example on page 4 "Histamine was not detectable in the vessel perfusates using a commercially available enzyme immunoassay with a detection limit 35 of 1.8 nM suggesting that H1R is activated agonist-independently by shear stress leading to Gq/11-protein activation and NO production thus resulting in flow-induced vasodilation."

We changed the manuscript by including more references and we tried to be more cautious with style and conclusions.

In the Results text for FRET studies it should be made more obvious at the outset which types of cells were used.

We have now denoted more clearly which type of cells we used (page 6, line 16f).

-Page 6 typo: ", but. Notably ..."

We have corrected this mistake.

Overall, a much more concise presentation of data and text would help the authors convince the reader of the main strengths of their case. Consideration should be given to focus on a smaller number of the most important findings.

We thank reviewer 1 for his advice and omitted all data regarding the role of PIP₂ and cholesterol for mechanosensation. We now focus on the finding that helix 8 is the essential structural element endowing GPCRs with mechanosensitivity. A simplified model on the role of H8 for mechanosensation is displayed in new Figure 6.

Reviewer #2 (Remarks to the Author):

The authors have performed a study showing that the C-terminal helix 8 (H8) provided mechanosensitivity to the histamine H1 GPCR. This makes this GPCR a new are endothelial sensor of shear stress in endothelial cells. The H8 might also be the link between the mechanosensitive GPCRs and the production of vasodilators by the endothelium.

The study is done properly using appropriate approaches. The data support the conclusion and the discussion is clear enough with reasonable arguments.

Nevertheless, some concerns remain.

1 – Figure 1: the authors should also show that mepyramine or desloratadine reduce flow-mediated nitrate production. Furthermore, it is also important to show the possible effect (or absence of effect) of the KO (and of mepyramine or desloratadine) of the contractility of the arteries and on the vasodilation induced by agonists (acetylcholine for example).

We appreciate the valuable recommendations of reviewer 2. In the whole manuscript, we changed the presentation of our data according to the Journals guidelines and we now display data as boxplot analysis plus interquartile range. We show individual data points if N number is ≤ 10 . Moreover, we applied iterative Grubb's test to identify and remove outliers. Significances were tested again after removal of the outliers. Moreover, colors were changed from Red and Green to Magenta and Blue to support colorblind readers.

We now provide additional data demonstrating that mepyramine and desloratadine significantly reduce flow-induced nitrate production similar to what we observed by analyzing H₁R gene-deficient arteries. We have included the new data in Figure 1J.

Moreover, we now show additional parameters of the analyzed vessels in Supplementary Figure 1: 1. Maximal vasoconstriction induced by 60 mM potassium chloride, which was used as a control at the beginning of each measurement prior to application of any blocker. Only vessel showing constriction of at least 30% were regarded as viable and were selected for further analysis.

2. Acetylcholine-induced vasodilations determined following the shear stress protocol.

3. Maximal vasodilations induced by application of Ca²⁺-free solution at intraluminal pressures of 50 mmHg.

4. Maximal vasodilations induced by application of Ca²⁺-free solution at intraluminal pressures of 120 mmHg.

No significant differences between wild-type vessels in the presence and absence of mepyramine or desloratadine and between different genotypes were observed. These findings suggest that all vessels were viable and comparable.

2 – Similarly, fig 1A; it would interesting to see the expression level of the other mechanosensitive GPCRs that you cite in the discussion (the AT1R for example)

In Figure 1A we now show mRNA expression levels of four other mechanosensitive GPCRs: AT₁R, PTH₁R, D₅R and GPR68. PTH₁R, D₅R and GPR68 were not expressed in HUVECs. However, we detected low mRNA levels of AT₁R in HUVECs that were about 400-fold lower than H₁R mRNA levels suggesting that AT₁R does not play a dominant role for mechanosensation in endothelial cells. Moreover, we analyzed new samples of HUVECs to complete the mRNA expression of some proteins (see Figure 1A and page 5, line 9ff).

3 – Discussion: link between cholesterol and mechanosensitivity: might more complexe in hypercholesterolemic patients. If I follow you rhyphotesis, their endothelium should be more sensitive to shear stress and then it should be better protected (shear increases eNOS level...etc).

We agree that the link between cholesterol and mechanosensitivity might be more complex and we can only speculate whether this has an impact on hypercholesterolemic patients. Since reviewer 4 advised us to omit all data concerning the role of PIP₂ and cholesterol on mechanosensation and reviewer 1 recommended us to concentrate on a smaller number of most important findings, we decided to delete the former Figure 6 and to omit all data and discussion concerning cholesterol and PIP₂ in the revised manuscript in agreement with the editor. We will analyze the role of cholesterol and PIP₂ for mechanosensation in more detail in a future study.

4 – Do you expect an equivalent H8 in AT1R or the other mechanosensitive GPCRs with the same role and pathway?

We appreciate the reviewer's recommendation to analyze the role of H8 for mechanosensation of AT₁Rs. To find out whether H8 is the essential structural element of AT₁Rs, too, we performed amino acid exchanges in H8¹⁹ from phenylalanine to proline (AT₁R-F309P and AT₁R-F309P-F131P) to disrupt the helix structure. Performing calcium imaging, all mutants were functional showing angiotensin II-induced calcium increases (Supplementary Figure 6H-6K). Hypoosmotically induced calcium increases were significantly reduced in the H8 mutants (Figure 4J and 4M and Supplementary Figure 6E-6G), indicating that H8 is also critical for mechanosensation of AT₁Rs.

5 – AT1R is expected to play a role in pressure-induced tone more than in flow-mediated relaxation. As both shear stress and hypo-osmotic shock activate the H8-associated pathway, do the authors expect a role for H8 in myogenic tone?

Since H8 is the critical structural element of AT₁Rs to sense mechanical stimuli, we speculate that H8 of AT₁R expressed in the vasculature of small resistance arteries is the essential structural motif for sensing mechanical forces and for converting them into vascular responses thereby contributing to the autoregulation of vessel tone^{5, 7, 14, 20}. We have integrated this statement in the discussion section (page 8, line 28f).

Reviewer #4:**Review report Erdogmus et al.**

In the present MS the authors present compelling evidence on the role of in the endothelial H1R receptor as a sensor of shear stress, mediating flow-induced vasodilation. Moreover, the mechanosensitivity can be measured as well in transfected HEK-293 cells. Using sophisticated single cell FRET measurements using conformational H1R biosensors, the authors show that the mechanosensitive response induces a different H1R conformation compared to the GPCR agonist histamine. Moreover, the authors present compelling evidence on the role of helix 8 in the mechanosensitive properties of the H1R and a number of other GPCRs and have tried to resolve the actual mechanism involved. Overall, a very comprehensive study with state-of-the art techniques that is revealing an very interesting feature of GPCR proteins.

Major comments

Abstract: in the last sentence the authors suggest that “H8 might display a therapeutic target”. I feel that that there is no actual foundation for this in the present work. There is no evidence that a small part of a GPCR can be seen as a specific target. Do the authors perhaps mean that it might constitute an interesting new region of the GPCR proteins, that could potentially be targeted as well? However, there is no evidence that this can be done without affect normal agonist function on the GPCR target or how one should try to get specificity for GPCRs with H8. I strongly advise to change the final sentence of the abstract (and also the final sentences in the discussion).

We thank reviewer 4 for his/her valuable comments.

In the whole manuscript, we changed the presentation of our data according to the Journals guidelines and we now display data as boxplot analysis plus interquartile range. We show individual data points if N number is ≤ 10 . Moreover, we applied Grubb's test to identify and remove outliers. Significances were tested again after removal of the outliers. Moreover, colors were changed from Red and Green to Magenta and Blue to support colorblind readers.

We appreciate the suggestions of reviewer 4 and we deleted the sentences in the abstract and in the discussion section.

Fig. 1B-E: Mepyramine is inhibiting the calcium responses to histamine and hypotonic solutions completely, but not the calcium responses to shear stress (1B. lower panel). What to make of this? Should one argue that in HUVECs the shear stress response is inducing calcium responses via a different mechanism than the hypotonic solution? Can the authors comment on this and mention something about these differences in the text?

Indeed our data suggest that the shear stress-induced calcium responses were significantly, but not fully blocked by mepyramine. The first calcium response induced by the switch from no flow to flow conditions caused the highest calcium transients, but should be regarded as a non-physiological stimulus, as endothelial cells are usually not exposed to no flow conditions. The second and more physiological response to increasing shear stress (from 4 to 20 dyn cm^{-2}) was also significantly, but not fully suppressed by mepyramine. The hypoosmotically induced calcium transients were nearly fully diminished by mepyramine. These discrepancies might be due to different mechanosensation and –transduction mechanisms that have evolved in endothelial cells and that are initiated by different stimuli. For example, laminar and oscillatory shear stress can cause distinct cellular responses (summarized in^{21, 22}). Thus, it is likely, that laminar shear stress and membrane stretch may induce different mechanosensitive signaling cascades. In endothelial cells, many proteins are involved in mechanoregulation. Among them are apical mechanosensors such as primary cilia, caveolae and the glycocalyx, but also GPCRs, receptor tyrosine kinases, ion channels, junctional mechanosensors such as platelet endothelial

cell adhesion molecule-1 (PECAM-1), VE-Cadherin and VEGF receptors and basal sensors such as integrins (summarized in²²). Therefore, it is likely that there is a complex regulation of mechanosignaling in endothelial cells and that different mechanical stimuli can induce different cellular responses. We have included these comments in the introduction (page 3, lines 24-33), in the results (page 5, line 14 and line 19) and in the discussion section (page 11, lines 15ff).

Fig. 2F: the authors show that the conformational changes of the H1R sensors after hypotonic solutions can only partially be inhibited by H1 blockers. As the H1R sensors still give good calcium responses (Fig.S2) upon histamine addition, it would be good to know if also a hypotonic solution is giving calcium responses and if that can be fully blocked by the used concentrations of mepyramine and levo-cetirizine.

We appreciate the reviewer's advice to perform additional calcium imaging experiments with HEK293 cells stably expressing the H₁R-FRET constructs H₁R-il3-b, H₁R-il3-e and H₁R-ct-b. We found that mepyramine and levocetirizine nearly fully suppressed hypoosmotically induced calcium signals (see new Supplementary Figure 4C-E) whereas the FRET signals were only partially inhibited. These findings suggest that the conformational changes we monitored in the presence of the inverse agonists using the H₁R-FRET sensors display inactive receptor conformations that do not allow for G-protein activation and subsequent calcium release. We have included these statements in the results section (page 7, lines 7ff).

Fig. 3B: next to the calcium measurements and the H1R sensors, the authors now introduce a third method (TRPC6 readout) to look at the functionality of the used H1R mutant. Since these mutants are already known to knock-out histamine functionality, I am not sure if this is very useful, especially as WT responses are not shown for comparison. I suggest to keep focus on the calcium tracings and the H1R sensors and omit the electrophysiology in this figure.

We thank the reviewer for this advice. However, in agreement with the editor we decided to keep the electrophysiological data and to add your recommended additional control experiments. We now show whole-cell measurements with wild-type H₁R and TRPC6 co-expressing HEK293 cells. Histamine, which was applied as a first stimulus, caused the largest TRPC6 current increases. Hypoosmotically induced membrane stretch and application of the TRPC6 channel activator OAG that was used as a control, also caused TRPC6 current increases that were not different from the currents monitored analyzing H₁R-mutant co-expressing cells (see new Figure 3C). Notably, TRPC6 channels *per se* are not mechanosensitive as previously demonstrated^{7, 23}.

Page 6: The authors also investigate if Gi/o and Gs coupled receptors are mechanosensitive, if the GPCR also contains H8. One important aspect that is however not shown is if these mechanosensitive responses can be counteracted by D2 or A2A antagonists. This is a relatively simple experiment that would strengthen the H1R case of blockade of mechanosensitive H1R signalling and shear-stress induced vasodilation by H1 antagonist.

We appreciate the reviewer's recommendation and we performed additional electrophysiological and FRET measurements. Analyzing CHO-K1 cells co-expressing Kir3.1/Kir3.2 channels and the D₂R, we found that the inverse agonist haloperidol fully suppressed hypoosmotically induced potassium currents. The initial application of dopamine served here as a control. Haloperidol even suppressed basal potassium currents suggesting a basal activity of D₂R. To compare mechanically induced potassium current increases elicited by a second stimulation with hypotonic solution, we performed control experiments applying dopamine as a first and hypotonic solution as a second stimulus (see new Figures 5B and 5C). Moreover, we performed FRET measurements with the A_{2A}R-FRET construct. Adenosine was applied as a positive control at the beginning of the measurements. In the presence of the selective inverse agonist ZM 241385 hypoosmotically induced FRET signals were significantly, but not fully suppressed (see new

Figures 5G and 5J). Altogether, we can now provide new data supporting the notion that inverse agonists can significantly suppress mechanically induced signaling.

Page 7: the part on the role of PIP2 is in my opinion the least developed part of the story. The part on wortmannin is questionable as at 20 μ M, wortmannin will hit many targets (see ChemBl or PubChem), so the effect cannot be attributed to PIP2 levels. I would omit this part. Next, potential binding partners for PIP2 binding are mutated. For the R mutations, data on membrane expression is missing and should be shown. Finally, some effects of cholesterol are reported on the effects of hypotonic solution on one FRET sensor, but a characterization of the effects of H1R agonist responses on FRET sensors or effects on H1R-mediated calcium responses are not shown. This part feels relatively incomplete and could be omitted from the paper in my opinion as well.

We agree with reviewer 4 that the part on the role of PIP₂ and cholesterol is not fully developed and requires further analysis. In agreement with the editor Dr. Mieck we decided to delete all data concerning PIP₂ and cholesterol regulation. We are planning to study the PIP₂ regulation in more detail in a future research study.

Discussion, page 8. In the discussion the authors get back to the original findings on the vasodilation effect after shear stress. The authors try to translate their results with cholesterol to a clinical setting, but for the studied mechanosensitive GPCRs no attention is given to the H1R. In contrast, the authors mention the AT1R instead. Yet, H1 antagonists have been used by millions of people and data on cardiovascular effects or risk factors will undoubtedly be available. In view of the present focus of the MS, I suggest that the authors pay some attention to this point.

We thank reviewer 4 for his/her advice. H₁R antagonists or inverse agonists are widely used as antihistaminic drugs. First generation antihistaminic drugs might cause dizziness and orthostatic hypotension due to their anti- α -adrenergic side effects. However, it is not documented that the H₁R selective second generation antihistaminic drugs significantly influence the blood pressure. Our findings suggest that antihistaminic drugs prevent flow-induced vasodilation but do not cause active vasoconstriction. Since regulation of vessel tone is vitally important, several mechanisms act in concert together to keep blood pressure constant and the signaling pathways that are involved might even vary between different vessel beds. Thus, dysregulation of blood pressure under antihistaminic medication might be masked. Interestingly, it was reported that the intake of antihistaminic drugs prior to sporting activity prevents post-exercise hypotension²⁴ and reduces blood flow²⁵ indicating that under certain conditions impaired flow-induced vasodilation by antihistaminic drugs might indeed influence blood pressure. We have included these statements in the discussion section (page 11, line 29ff).

Minor comments

Page 6, line 1: *mechanicallyinduced* should be *mechanically induced*

We have corrected this typing error.

Page 6, line 11: incomplete sentence

We have corrected the sentence.

Page 7, line 37: I do not think that the terminology of “gene-deficient mouse” is appropriate?

We have changed the sentence to “We used pharmacological tools as well as knock-out ($H_1R^{-/-}$ and $H_{1/2/3/4}R^{-/-}$) and smooth muscle specific inducible knock-down ($SmG_{q/11}^{-/-}$) mice...” in the discussion section (page 10, line 5f).

References:

1. Wang S, Chennupati R, Kaur H, Iring A, Wettschureck N, Offermanns S. Endothelial cation channel PIEZO1 controls blood pressure by mediating flow-induced ATP release. *J Clin Invest* **126**, 4527-4536 (2016).
2. Iring A, *et al.* Shear stress-induced endothelial adrenomedullin signaling regulates vascular tone and blood pressure. *J Clin Invest* **129**, 2775-2791 (2019).
3. Dela Paz NG, Frangos JA. Rapid flow-induced activation of Galphaq/11 is independent of Piezo1 activation. *Am J Physiol Cell Physiol* **316**, C741-C752 (2019).
4. Mederos y Schnitzler M, Storch U, Gudermann T. Mechanosensitive Gq/11 protein-coupled receptors mediate myogenic vasoconstriction. *Microcirculation* **23**, 621-625 (2016).
5. Storch U, Blodow S, Gudermann T, Mederos y Schnitzler M. Cysteinyl leukotriene 1 receptors as novel mechanosensors mediating myogenic tone together with angiotensin II type 1 receptors. *Arterioscler Thromb Vasc Biol* **35**, 121-126 (2015).
6. Forst AL, *et al.* Podocyte purinergic P2X4 channels are mechanotransducers that mediate cytoskeletal disorganization. *J Am Soc Nephrol* **27**, 848-862 (2016).
7. Mederos y Schnitzler M, *et al.* Gq-coupled receptors as mechanosensors mediating myogenic vasoconstriction. *EMBO J* **27**, 3092-3103 (2008).
8. Yamamoto K, Ando J. Endothelial cell and model membranes respond to shear stress by rapidly decreasing the order of their lipid phases. *J Cell Sci* **126**, 1227-1234 (2013).
9. Yamamoto K, Ando J. Vascular endothelial cell membranes differentiate between stretch and shear stress through transitions in their lipid phases. *Am J Physiol Heart Circ Physiol* **309**, H1178-1185 (2015).
10. Cantor RS. Lateral pressures in cell membranes: A mechanism for modulation of protein function. *Journal of Physical Chemistry B* **101**, 1723-1725 (1997).
11. Lipowsky HH, Kovalcheck S, Zweifach BW. The distribution of blood rheological parameters in the microvasculature of cat mesentery. *Circ Res* **43**, 738-749 (1978).
12. Papaioannou TG, Stefanadis C. Vascular wall shear stress: basic principles and methods. *Hellenic J Cardiol* **46**, 9-15 (2005).
13. Storch U, Mederos y Schnitzler M, Gudermann T. G protein-mediated stretch reception. *Am J Physiol Heart Circ Physiol* **302**, H1241-1249 (2012).
14. Blodow S, *et al.* Novel role of mechanosensitive AT1B receptors in myogenic vasoconstriction. *Pflugers Arch* **466**, 1343-1353 (2014).
15. Welsh DG, Morielli AD, Nelson MT, Brayden JE. Transient receptor potential channels regulate myogenic tone of resistance arteries. *Circ Res* **90**, 248-250 (2002).

16. Earley S, Heppner TJ, Nelson MT, Brayden JE. TRPV4 forms a novel Ca²⁺ signaling complex with ryanodine receptors and BKCa channels. *Circ Res* **97**, 1270-1279 (2005).
17. Bjorling K, *et al.* Role of age, Rho-kinase 2 expression, and G protein-mediated signaling in the myogenic response in mouse small mesenteric arteries. *Physiol Rep* **6**, e13863 (2018).
18. Delaey C, Van de Voorde J. Pressure-induced myogenic responses in isolated bovine retinal arteries. *Investigative Ophthalmology & Visual Science* **41**, 1871-1875 (2000).
19. Zhang H, *et al.* Structure of the Angiotensin receptor revealed by serial femtosecond crystallography. *Cell* **161**, 833-844 (2015).
20. Pires PW, Ko EA, Pritchard HAT, Rudokas M, Yamasaki E, Earley S. The angiotensin II receptor type 1b is the primary sensor of intraluminal pressure in cerebral artery smooth muscle cells. *J Physiol* **595**, 4735-4753 (2017).
21. Johnson BD, Mather KJ, Wallace JP. Mechanotransduction of shear in the endothelium: basic studies and clinical implications. *Vasc Med* **16**, 365-377 (2011).
22. Givens C, Tzima E. Endothelial mechanosignaling: Does one sensor fit all? *Antioxid Redox Signal* **25**, 373-388 (2016).
23. Gottlieb P, *et al.* Revisiting TRPC1 and TRPC6 mechanosensitivity. *Pflugers Arch* **455**, 1097-1103 (2008).
24. Lockwood JM, Wilkins BW, Halliwill JR. H1 receptor-mediated vasodilatation contributes to postexercise hypotension. *J Physiol* **563**, 633-642 (2005).
25. Emhoff CAW, Barrett-O'Keefe Z, Padgett RC, Hawn JA, Halliwill JR. Histamine-receptor blockade reduces blood flow but not muscle glucose uptake during postexercise recovery in humans. *Experimental Physiology* **96**, 664-673 (2011).

Reviewers' Comments:

Reviewer #1:

Remarks to the Author:

Concern remains about the extensive use of hypotonic shock as the stimulus for activating the receptors. This is because the aim of the study was to advance understanding of responses to mechanical force. The authors employed severe shock, caused by approximate halving of the tonicity of the medium. Such a treatment of cells obviously exposes the cells to a big change in tonicity. Therefore effects could have been in response to this change in tonicity. The authors also used exclusion/inclusion of a high concentration of mannitol to cause the change in tonicity. Mannitol is an antioxidant. Therefore effects could have been in response to changes in the availability of reactive oxygen species.

The authors refer to their stimulus protocol as "hyposmotic membrane stretch". However they do not show that the stimulus actually caused membrane stretch and do not explore whether the effects could have been due to tonicity or reactive oxygen changes. It is possible that there was no membrane stretch if these cells were able to avoid cell volume changes through homeostatic mechanisms.

Retention of this technical approach is a concern. It affects large elements of the study and its conclusions.

Reviewer #2:

Remarks to the Author:

No further comment

Reviewer #3:

Remarks to the Author:

I think the authors have written a good rebuttal and have substantially improved their paper, both in writing and with new experimental data (and by omitting more speculative data on e.g. PIP2). I am very pleased with their rebuttal and changes following my review.

Despite the ongoing technical discussion on the hypotonic shock paradigm with rev. 1., I think the authors make a strong case that will certainly impact the GPCR-field. As such, I would advise to accept the paper in the present form, as the conclusions are not solely based on the hypotonic shock paradigm.

Point-by-point response to the reviewers

REVIEWERS' COMMENTS:

Reviewer #1 (Remarks to the Author):

Concern remains about the extensive use of hypotonic shock as the stimulus for activating the receptors. This is because the aim of the study was to advance understanding of responses to mechanical force. The authors employed severe shock, caused by approximate halving of the tonicity of the medium. Such a treatment of cells obviously exposes the cells to a big change in tonicity. Therefore effects could have been in response to this change in tonicity. The authors also used exclusion/inclusion of a high concentration of mannitol to cause the change in tonicity. Mannitol is an antioxidant. Therefore effects could have been in response to changes in the availability of reactive oxygen species.

The authors refer to their stimulus protocol as “hypoosmotic membrane stretch”. However they do not show that the stimulus actually caused membrane stretch and do not explore whether the effects could have been due to tonicity or reactive oxygen changes. It is possible that there was no membrane stretch if these cells were able to avoid cell volume changes through homeostatic mechanisms.

Retention of this technical approach is a concern. It affects large elements of the study and its conclusions.

We agree with reviewer 1 that hypoosmotically induced membrane stretch is not a physiological stimulus. However, our previous findings showed that hypoosmotic stimulation elicited the same effects as direct membrane stretch^{1,2}. Thus, in this study, we used this stimulus as a fast screening method and additionally applied fluid shear stress as a physiological stimulus.

Notably, hypoosmotic stimulation is widely used as a mechanical stimulus that causes membrane stretch^{3, 4, 5, 6, 7, 8, 9, 10, 11, 12}. To prevent confounding effects of unwanted changes in ion concentrations, hypoosmotic solutions were applied that had the same ion concentrations as the isosmotic solutions. Hereby, the isosmotic solution contains the same ions at the same concentration as the hypoosmotic solution and is additionally supplemented with the inert and non-metabolizable sugar alcohol mannitol to adjust osmolarity to about 300 mOsm kg⁻¹. Thus, changes in the ionic strength of both solutions are prevented to preclude shifts of the equilibrium potentials of the ions and subsequent changes of the membrane potential^{6, 8, 9, 10, 11}.

Recently, applying super-resolution techniques and atomic force microscopy, cell membrane stretch of cells in response to hypoosmotic stress was exactly visualized^{4, 13, 14}. In our hands, short time stimulation with hypoosmotic solution (150 mOsm kg⁻¹) is a strong but innocuous stimulus which is reversible and repeatable. Notably, hypoosmotic stress of 150 mOsm kg⁻¹ results in small volume increases and resultant quantifiable membrane stretch which was regarded as a mild osmotic stimulus^{3,4}.

Mannitol is often used to balance out osmolality, because it is well water soluble, membrane impermeable and non-metabolizable. We agree with reviewer 1 that mannitol has antioxidative affects similar to other sugar alcohols that protect cells against oxidative stress by scavenging reactive oxygen species.

However, we were unable to detect any evidence in the scientific literature supporting the notion that extracellular reactive oxygen species might be involved in ligand-independent activation of H₁Rs in particular or of GPCRs in general. On the contrary, it is well known that in neutrophils

H₁Rs can elicit oxidative burst and intracellular ROS formation downstream of H₁R activation (summarized in ¹⁵). Thus, the concept of ligand-independent activation of H₁Rs by mannitol-induced reduction of extracellular ROS is purely speculative and not supported by scientific evidence.

Since effects of mannitol on GPCR activation cannot be fully excluded, we have included the following statement in the discussion section: “This approach has its limitations, because in the vasculature, a rapid decrease of tonicity cannot be regarded as a physiological stimulus and the supplemented sugar alcohol mannitol in the isosmotic bath solution might exert additional antioxidative effects.” (page 11, line 13ff).

Reviewer #2 (Remarks to the Author):

No further comment

We thank reviewer 2 for his/her contribution to the peer review of our manuscript

Reviewer #3 (Remarks to the Author):

I think the authors have written a good rebuttal and have substantially improved their paper, both in writing and with new experimental data (and by omitting more speculative data on e.g. PIP2). I am very pleased with their rebuttal and changes following my review.

Despite the ongoing technical discussion on the hypotonic shock paradigm with rev. 1., I think the authors make a strong case that will certainly impact the GPCR-field. As such, I would advise to accept the paper in the present form, as the conclusions are not solely based on the hypotonic shock paradigm.

We thank reviewer 3 for his/her constructive criticism, which helped to improve the manuscript.

References:

1. Mederos y Schnitzler M, *et al.* Gq-coupled receptors as mechanosensors mediating myogenic vasoconstriction. *EMBO J* **27**, 3092-3103 (2008).
2. Forst AL, *et al.* Podocyte Purinergic P2X4 Channels Are Mechanotransducers That Mediate Cytoskeletal Disorganization. *J Am Soc Nephrol* **27**, 848-862 (2016).
3. Guo Y, Yang L, Haught K, Scarlata S. Osmotic Stress Reduces Ca²⁺ Signals through Deformation of Caveolae. *J Biol Chem* **290**, 16698-16707 (2015).
4. Yang L, Scarlata S. Super-resolution Visualization of Caveola Deformation in Response to Osmotic Stress. *J Biol Chem* **292**, 3779-3788 (2017).
5. Stuhlmann T, Planells-Cases R, Jentsch TJ. LRRC8/VRAC anion channels enhance beta-cell glucose sensing and insulin secretion. *Nat Commun* **9**, 1974 (2018).
6. Hennes A, *et al.* Functional expression of the mechanosensitive PIEZO1 channel in primary endometrial epithelial cells and endometrial organoids. *Sci Rep* **9**, 1779 (2019).
7. Planells-Cases R, *et al.* Subunit composition of VRAC channels determines substrate specificity and cellular resistance to Pt-based anti-cancer drugs. *EMBO J* **34**, 2993-3008 (2015).

8. Alessandri-Haber N, *et al.* Hypotonicity induces TRPV4-mediated nociception in rat. *Neuron* **39**, 497-511 (2003).
9. Vriens J, Watanabe H, Janssens A, Droogmans G, Voets T, Nilius B. Cell swelling, heat, and chemical agonists use distinct pathways for the activation of the cation channel TRPV4. *Proceedings of the National Academy of Sciences of the United States of America* **101**, 396-401 (2004).
10. Gomis A, Soriano S, Belmonte C, Viana F. Hypoosmotic- and pressure-induced membrane stretch activate TRPC5 channels. *J Physiol* **586**, 5633-5649 (2008).
11. Liedtke W, *et al.* Vanilloid receptor-related osmotically activated channel (VR-OAC), a candidate vertebrate osmoreceptor. *Cell* **103**, 525-535 (2000).
12. Hong K, Li M, Nourian Z, Meininger GA, Hill MA. Angiotensin II Type 1 Receptor Mechanoactivation Involves RGS5 (Regulator of G Protein Signaling 5) in Skeletal Muscle Arteries: Impaired Trafficking of RGS5 in Hypertension. *Hypertension* **70**, 1264-1272 (2017).
13. Spagnoli C, Beyder A, Besch S, Sachs F. Atomic force microscopy analysis of cell volume regulation. *Phys Rev E Stat Nonlin Soft Matter Phys* **78**, 031916 (2008).
14. Jaiswal A, Hoerth CH, Zuniga Pereira AM, Lorenz H. Improved spatial resolution by induced live cell and organelle swelling in hypotonic solutions. *Sci Rep* **9**, 12911 (2019).
15. Ciz M, Lojek A. Modulation of neutrophil oxidative burst via histamine receptors. *Br J Pharmacol* **170**, 17-22 (2013).